# A HIMU-like component in Mariana Convergent Margin magma sources during initial arc rifting revealed by melt inclusions

Xiaohui Li[1,2,3], Osamu Ishizuka[4,5], Robert J. Stern [6], Sanzhong Li [1,2] ✉,
Zhiqing Lai[1], Ian Somerville[7], Yanhui Suo[1,2], Long Chen [1,2] & Hongxia Yu[8]

Compositions of island arc and back-arc basin basalts are often used to trace the recycling of subducted materials. However, the contribution of subducted components to the mantle source during initial arc rifting before back-arc basin spreading is not yet well constrained. The northernmost Mariana arc is ideal for studying this because the transition from rifting to back-arc spreading is happening here. Here we report major and trace element and Pb isotopic compositions of olivine-hosted melt inclusions from lavas erupted during initial rifting at 24°N (NSP-24) and compare them with those in active arc front at 21°N and mature back-arc basin at 18°N. NSP-24 high-K melt inclusions have highly radiogenic Pb compositions and are close to those of the HIMU end-member, suggesting the presence of this component in the magma source. The HIMU-like component may be stored in the over-riding plate and released into arc magma with rifting. HIMU-type seamounts may be subducted else-where beneath the Mariana arc, but obvious HIMU-type components appear only in the initial stages of arc rifting due to the low melting degree and being consumed during the process of back-arc spreading.

Subduction zones are important places for chemical recycling in the Earth[1]. Oceanic island arcs, where continental crust is absent, are key locations for identifying various subduction components and recycling processes[2]. Arc magmas are characterized by the enrichment of large-ion lithophile elements (LILE) and light rare-earth elements (LREE), relative to mid-ocean ridge basalts (MORB), which is mainly attributed to the dehydration and melting of subducted components, including sediments, altered oceanic crust (AOC)[3–5] and seamounts[6–8]. The Izu-Bonin-Mariana (IBM) arc is a typical oceanic island arc, where sediment, AOC, and seamounts of the Pacific Plate subduct, making it

an ideal system to study subducted component influences on magma genesis[4,9–12]. Geochemical studies of IBM lavas show that the contribution of subduction components varies along the length of the arc[12]. In the Izu-Bonin arc, hydrous fluid released from pelagic sediment and AOC dominate contributions to the mantle wedge, while in the northern Mariana arc, there is a greater contribution of sediment melt[4,10,12]. The mantle source beneath the Mariana back-arc basin gradually changes from subduction-modified to MORB-like mantle with decreasing latitude and basin widening[13–16]. In addition, a few arc volcanoes (Iwo Jima and Kita Iwo Jima) in the southern Izu-Bonin arc have

[1]Frontiers Science Center for Deep Ocean Multispheres and Earth System, Key Lab of Submarine Geosciences and Prospecting Techniques, MOE and College of Marine Geosciences, Ocean University of China, Qingdao 266100, China. [2]Laboratory for Marine Mineral Resources, Qingdao Marine Science and Technology Center, Qingdao 266237, China. [3]Key Laboratory of Marine Geology and Environment, Institute of Oceanology, Chinese Academy of Sciences, Qingdao 266071, China. [4]Institute of Geology and Geoinformation, Geological Survey of Japan/AIST, Central 7, 1-1-1, Higashi, Tsukuba, Ibaraki 305-8567, Japan. [5]Japan Agency for Marine-Earth Science and Technology, 2-15 Natsushima, Yokosuka, Kanagawa 237-0061, Japan. [6]Department of Sustainable Earth Systems Science, University of Texas at Dallas, Richardson, TX 75080, USA. [7]UCD School of Earth Sciences, University College Dublin, Belfield, Dublin 4, Ireland. [8]Guangxi Key Laboratory of Hidden Metallic Ore Deposits Exploration, Guilin University of Technology, Guilin 541006, China. ✉e-mail: sanzhong@ouc.edu.cn

lavas with highly radiogenic Pb compositions ($^{206}Pb/^{204}Pb > 19.4$), reflecting the result of subducted seamounts with HIMU (high-μ or high time-integrated $^{238}U/^{204}Pb$) characteristics[10,12,17].

The North Northern Seamount Province (N-NSP) of the Mariana arc between 23°N and 24°N is immediately north of where the Mariana Trough terminates (Fig. 1) and is undergoing northward-propagating back-arc rifting, but seafloor spreading to form oceanic crust has not yet begun. Stern et al.[18] suggested that magmas that erupted in this location just prior to initial arc rifting have different mantle source compositions than normal arc magmas. High-precision Pb isotopic studies of whole-rocks show that the compositions of the mantle source beneath this area are complex, involving subducted pelagic sediment, AOC and HIMU-like Cretaceous seamounts, and volcaniclastics[10,19]. Compared to whole-rocks, olivine-hosted melt inclusions are less affected by later magmatic processes such as magma mixing and fractional crystallization, because they are protected by their host minerals, thus retaining more complete

compositional information of the primitive magma and the mantle source[20–23]. In addition, melt inclusions also record information about magmatic evolution prior to eruption[20], which can help us better understand how subduction components migrate through the mantle wedge. Therefore, in tandem with studying magmatic products during arc rifting, identifying the composition of mantle sources before rifting begins allows us to better understand lithospheric processes at the initiation of rifting, and sheds light on the various subduction components[19]. In this paper, we report major and trace elements and Pb isotopic analyses of olivine-hosted melt inclusions from basalts at three sites in the northern Mariana arc. These include samples collected from the eastern scarp of the remnant arc West Mariana Ridge (WMR) which is the conjugate pair of the northernmost North Mariana Seamount Province around 24°N (NSP-24; Fig. 1), the Mariana arc at 21°N (NSP-21) and the back-arc basin at 18°N (MT-18). We use these data to identify trace element and Pb isotopic compositions of mantle sources, compare and analyze their spatial and temporal variabilities at

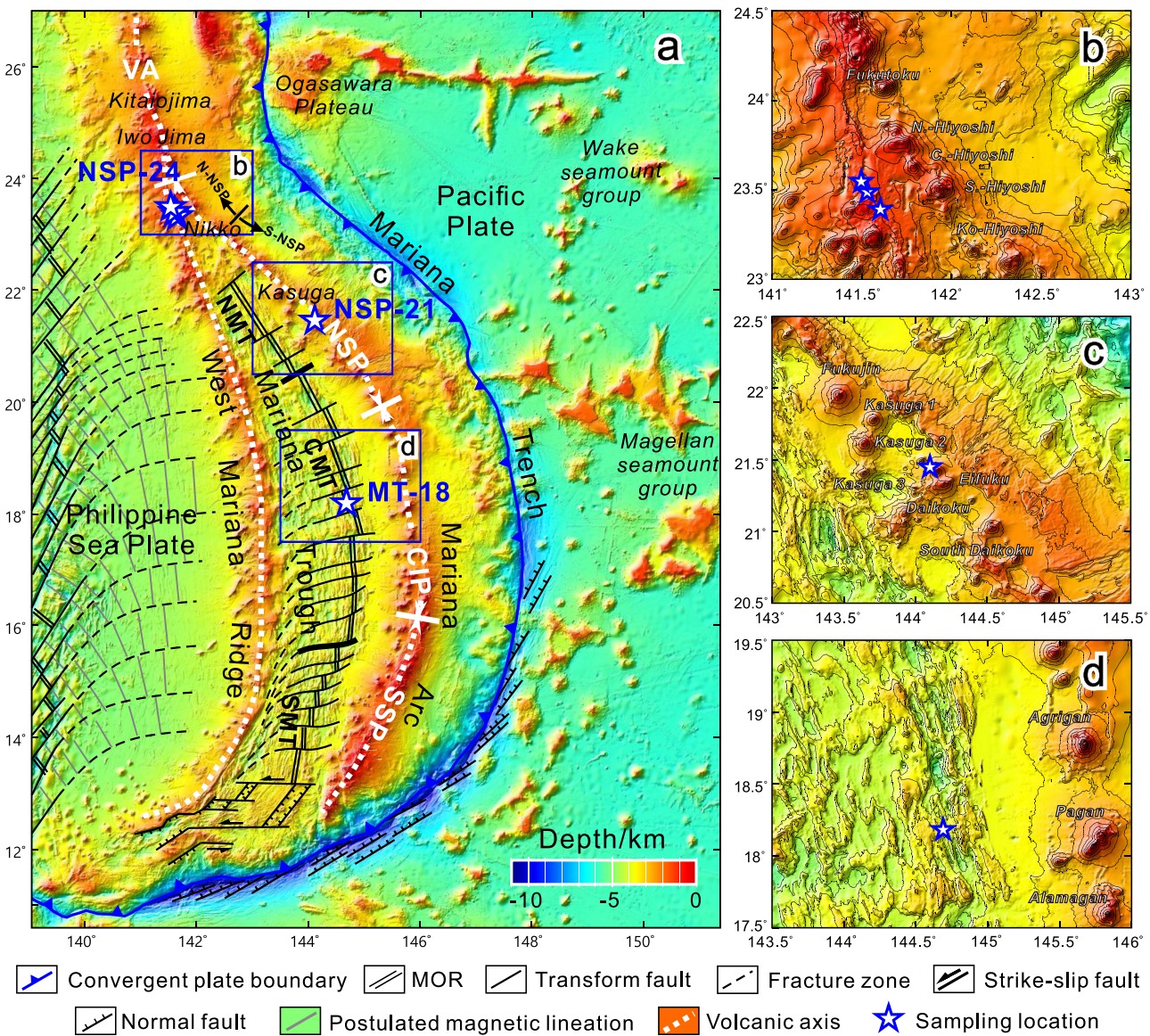

**Fig. 1 | Regional map of the Mariana convergent margin.** Map of the Mariana subduction zone showing the locations of the studied samples (**a**). The base map data is from SRTM15 + V2.5.5[93]. Previously studied whole-rock samples of the corresponding locations and the detailed sample information are shown in Supplementary Data 1. Insets show detailed locations of samples from the NSP-24 (**b**), NSP-21 (**c**) and MT-18 (**d**). MOR - Mid-Ocean Ridge; NSP - Northern Seamount Province; CIP - Central Islands Province; SSP - Southern Seamount Province; NMT - Northern Mariana Trough; CMT - Central Mariana Trough; SMT - Southern Mariana Trough.

different tectonic evolutionary stages, and illuminate the causes controlling these variations.

The IBM convergent margin marks the eastern edge of the Philippine Sea Plate and formed in response to the westward subduction of the Pacific Plate (Fig. 1). The Mariana Arc and Mariana Trough back-arc basin are different but related magma-tectonic systems. The magmatic arc is subdivided into four segments, from north to south: the southern Volcano Arc (VA, 24–28°N), the Northern Seamount Province (NSP, 20.7–24°N), the Central Islands Province (CIP, 16–20.7°N), and the Southern Seamount Province (SSP, south of 16°N)[11,24]. The Mariana Trough is subdivided into the Northern Mariana Trough (NMT, 21–24°N), the Central Mariana Trough (CMT, 17.6–21°N), and the Southern Mariana Trough (SMT, 12.5–17.5°N)[25,26]. The Mariana Trough south of 19°15′N is dominated by seafloor spreading which is propagating northward into the NMT, which is characterized by rifting[14,27,28]. We focus our study on the CMT and NSP (Fig. 1), which have different mantle sources affected by different proportions and types of subduction components. The CMT includes a mature back-arc spreading ridge with spreading rates ranging from 3.0 to 4.4 cm/a[29,30]; volcanism occurs near the main spreading axis[27,31]. The CMT has a maximum width of ~250 km at 18° N and back-arc spreading started at about 6 Ma[32,33]. CMT crustal structure is similar to that of a slow-spreading mid-oceanic ridge; seismic profiles show that this crust is about 6 km thick[11]. CMT lavas have geochemical and isotopic characteristics of MORB with minor involvement of subduction components[14,28,34–37]. The depth to the subducted slab beneath Mariana arc volcanoes along the magmatic front is ~130 km[38]. The NSP can be further subdivided into north NSP (N-NSP, 23–24°N) and south NSP (S-NSP, 20.7–23°N) reflecting differences in enriched components in the magma source[25,26,39,40] (Fig. 1). The N-NSP is where the propagating Mariana Trough rift intersects the Mariana arc, and the WMR (remnant arc) terminates here[19,41]. The NMT extensional zone terminates at Nikko volcano (23.1°N)[14]. The northward termination of the Mariana Trough may also have been caused by the ongoing collision of the Ogasawara Plateau with the IBM arc at 26 °N[42] (Fig. 1).

N-NSP and southern VA arc lavas are shoshonitic with distinct enrichments in incompatible major and trace elements and radiogenic isotopes; this arc segment is also called the Alkalic Volcano Province (AVP)[26]. In contrast, S-NSP lavas are not as enriched in incompatible trace elements as those of the N-NSP[10,26,40,43]. This study focuses on samples from around 18°N in the CMT, 21°N in the S-NSP, and 24°N in the WMR near the N-NSP (hereafter MT-18, NSP-21, and NSP-24, respectively) (Fig. 1). These correspond to magmatism associated with back-arc spreading, active arc, and arc rifting, respectively[14]. All studied samples are basalts, with olivine, clinopyroxene, and plagioclase as the main phenocrysts (Supplementary Fig. 1 and Supplementary Data 1, 2). Olivine phenocrysts in all samples are euhedral to subhedral and show no obvious compositional zonation (Supplementary Fig. 1). Round or elliptical melt inclusions 20–60 μm across are randomly distributed in some olivine phenocrysts (Supplementary Fig. 1), indicating a primary origin[44]. Compositions of all studied samples plot within the range of the previous whole-rock data (Supplementary Figs. 2–4), suggesting that they are representative.

## Results and Discussion
### Chemical compositions of host olivines and melt inclusions
The chemical compositions of host olivines and melt inclusions are presented in Supplementary Data 3. Olivine compositions vary significantly in different sampling areas, with Forsterite contents [Fo, atomic 100×Mg/(Mg + Fe)] decreasing from NSP-24 (83.8–90.7) to MT-18 (76.6–87.8) to NSP-21 (70.8–78.5). Almost all host olivines have high CaO contents (average = 0.18 wt.%, $n = 300$), indicating that they crystallized from magma, rather than being mantle-derived xenocrysts (CaO <0.1 wt.%[45]).

Melt inclusion compositions may be modified by crystallization and/or re-equilibration after being entrapped by their host minerals, i.e., post-entrapment crystallization (PEC)[20,46,47]. The measured major element compositions of melt inclusions were corrected for PEC Fe-loss using the method described by ref. 46 and recalculated with PETROLOG 3 software[47]. Olivine-melt equilibrium was set by applying the model of ref. 48. The initial FeOt content of the trapped melt can be obtained by using the FeOt fractionation trend of whole-rock samples[20,46], which were assumed to be 8.0 wt.% in MT-18, 9.0 wt.% in NSP-21 and 9.0 wt.% in NSP-24[26,49]. These corrections mainly affect MgO and FeO concentrations but have little effect on other elements, thus the overall geochemical trends and composition changes remain unaffected. We focus on the ratios of incompatible trace elements that are not significantly affected by PEC Fe-loss.

Compositions of melt inclusions after correcting for PEC Fe-loss are listed in Supplementary Data 3 and displayed on the total alkali-silica (TAS) diagram. On the TAS diagram, melt inclusions show larger compositional variation than their host whole-rocks (Supplementary Fig. 2). Most MT-18 and NSP-21 melt inclusions have compositions similar to those of their host whole-rocks, whereas some NSP-24 melt inclusion compositions are more primitive (Supplementary Fig. 2). MT-18 melt inclusions are subalkaline, low-K tholeiites. Some NSP-21 melt inclusions have high enough alkali contents to be classified as alkaline series; these are the first alkaline compositions reported from the S-NSP. According to their alkali contents, we identified two groups of both NSP-21 and NSP-24 melt inclusions. NSP-21 and NSP-24 group 1 melt inclusions plot in the medium-K calc-alkaline series field (Supplementary Fig. 2b). NSP-24 group 2 melt inclusions plot in the high-K calc-alkaline series field, whereas NSP-21 group 2 melt inclusions plot in high-K calc-alkaline to shoshonitic fields (Supplementary Fig. 2b). The major oxides versus MgO contents of different group melt inclusions do not covary systematically (Supplementary Fig. 3), indicating that fractional crystallization was not the only cause of magmatic variability but that different melt sources were also important.

In primitive mantle-normalized trace element "spider" diagrams, MT-18 melt inclusions are similar to E-MORB, except for slightly depleted Nb and Ta and enriched Pb contents (Fig. 2a). In contrast, NSP-21 and NSP-24 melt inclusions are markedly enriched in LILEs and REEs and depleted in high-field-strength elements (HFSEs), exhibiting arc-like trace element distribution patterns (Fig. 2c, e). NSP-21 group 1 melt inclusions are chemically similar to their host whole-rocks (Fig. 2 and Supplementary Fig. 5), whereas group 2 melt inclusions are strongly enriched in Li, Rb, Ba, and K but depleted in Cu, Zn, and Pb (Fig. 2 and Supplementary Fig. 4). NSP-24 group 1 melt inclusions also have similar trace element distribution patterns to their host whole-rocks and other AVP basaltic lavas (e.g., North Hiyosh), but incompatible trace element contents (e.g. Rb, K, P) of group 2 melt inclusions are higher (Fig. 2 and Supplementary Fig. 4, 5). Chondrite-normalized REE patterns of MT-18 melt inclusions are similar to those of E-MORB, showing little fractionation of LREEs and HREEs (Fig. 2b, d, f). In contrast, NSP-21 and NSP-24 melt inclusions show similar strong fractionation of LREEs from HREEs (Fig. 2d, f). NSP-24 group 2 melt inclusions have higher LREE contents than NSP-24 group 1 melt inclusions (Fig. 2f).

### Pb isotope compositions of melt inclusions
The Pb isotopic compositions of olivine-hosted melt inclusions are listed in Supplementary Data 3. Each of the five types of melt inclusions defined above has a narrow range of Pb isotope compositions. The total range that encompasses most of the global variation seen in oceanic basalts and that is far beyond the host whole-rocks (Fig. 3a). There is no significant relationship between Pb isotope variations of different group melt inclusions and the extent of magmatic fractionation as captured by MgO contents or by the Fo content of host olivines (Supplementary Fig. 6).

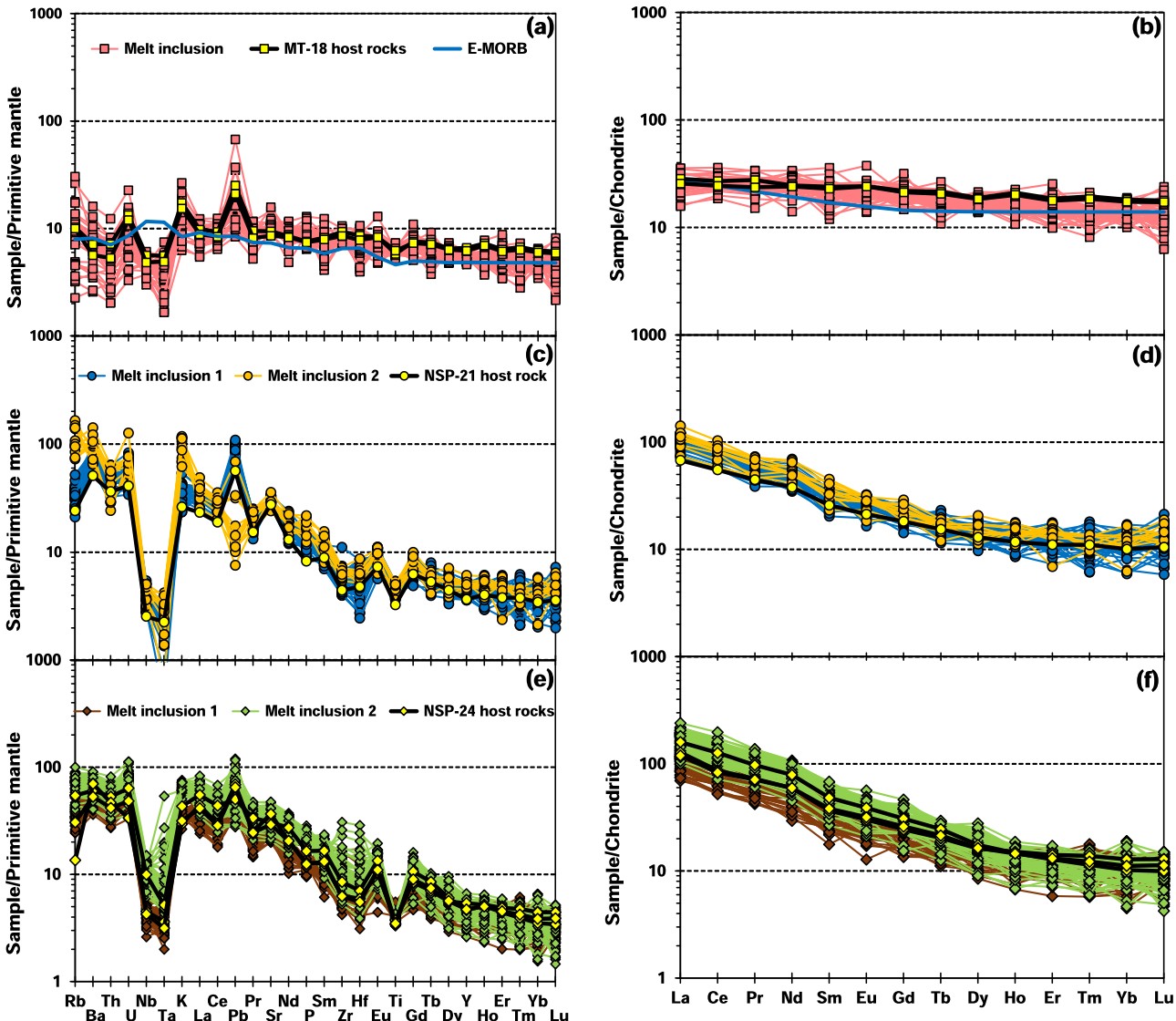

**Fig. 2 | Trace element compositions of samples investigated in this study.** Trace element patterns normalized to primitive mantle concentrations (**a**, **c**, **e**), and rare earth element patterns normalized to chondritic values (**b**, **d**, **f**) for melt inclusions and their host whole rocks. Normalized-primitive mantle and -chondrite values are from ref. [94]. There are two different groups of melt inclusions in NSP-21 and NSP-24, respectively.

The $^{207}Pb/^{206}Pb$ and $^{208}Pb/^{206}Pb$ values of MT-18 melt inclusions range from 0.8344 to 0.8612 and 2.0607 to 2.1145, with mean values of $0.8467 \pm 0.0117$ (2 SD, $n = 34$) and $2.0809 \pm 0.0268$ (2 SD, $n = 34$), respectively. These values all plot within the Indian Ocean-type mantle field (Fig. 3a). Actually, the radiogenic isotope composition of the mantle wedge is very commonly observed to be similar to Indian Ocean-type mantle for almost all Western Pacific Subduction zones (e.g., Kamchatka, Kuriles, IBM, Ryukyu)[50–54]. The $^{207}Pb/^{206}Pb$ and $^{208}Pb/^{206}Pb$ ratios of NSP-21 melt inclusions range from 0.8223 to 0.8780 and 2.0380 to 2.1503. NSP-21 group 2 melt inclusions have more old uranogenic and more thorogenic Pb (average $^{207}Pb/^{206}Pb = 0.8653 \pm 0.0215$, $^{208}Pb/^{206}Pb = 2.1243 \pm 0.0494$, 2 SD, $n = 11$) than group 1 melt inclusions (average $^{207}Pb/^{206}Pb = 0.8253 \pm 0.0041$, $^{208}Pb/^{206}Pb = 2.0430 \pm 0.0069$, 2 SD, $n = 31$) and tend towards the enriched mantle type I (EM I) field (Fig. 3a). The two groups of NSP-24 melt inclusions also have very different Pb isotopic compositions. Group 1 melt inclusions have a range of $^{207}Pb/^{206}Pb$ and $^{208}Pb/^{206}Pb$ ratios from 0.8202 to 0.8300 and 2.0343 to 2.0508 with mean values of $0.8246 \pm 0.0054$ (2 SD, $n = 24$) and $2.0433 \pm 0.0096$ (2 SD, $n = 24$), respectively. These values fall within the range of previously studied NSP-24 whole-rocks (Fig. 3a). NSP-24 group 1

melt inclusion Pb isotopes are remarkably similar to NSP-21 group 1 melt inclusions. The $^{207}Pb/^{206}Pb$ and $^{208}Pb/^{206}Pb$ ratios of NSP-24 group 2 melt inclusions are lower than those of NSP-24 group 1 melt inclusions, which range from 0.7979 to 0.8078 and 2.0062 to 2.0253, with mean values of $0.8028 \pm 0.0043$ (2 SD, $n = 33$) and $2.0119 \pm 0.0087$ (2 SD, $n = 33$), respectively. Pb isotopic compositions of NSP-24 group 2 melt inclusions fall outside the Pacific- and Indian Ocean-type mantle range and trend towards the HIMU end-member (Fig. 3a).

## Influence of fractional crystallization and partial melting on melt inclusion elemental variations

Trace element variations between the different groups of melt inclusions in NSP-21 and NSP-24 might result from the fractional crystallization of minerals and partial melting of the mantle source. Although the MgO contents of different group melt inclusions vary, they do not differ significantly in the major minerals likely to be involved in fractional crystallization (such as olivine, pyroxene, and plagioclase) (Supplementary Fig. 3). Nor can fractional crystallization explain the large and systematic variations in Pb isotopic compositions seen between group 1, NSP-21 group 2 and NSP-24 group 2 melt inclusions.

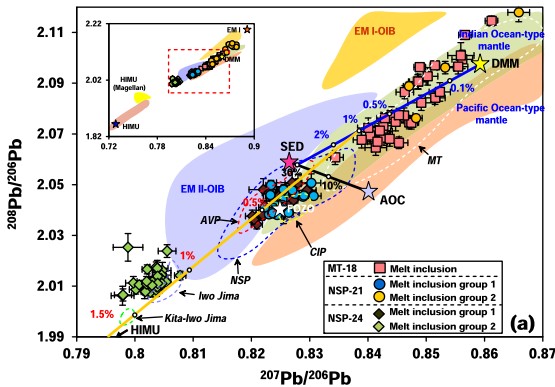
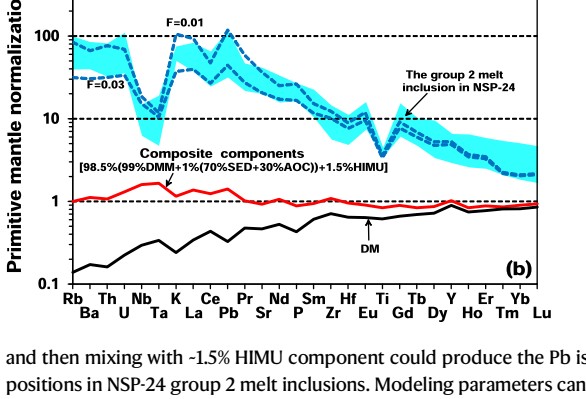

**Fig. 3 | Simulation of trace elements and Pb isotopic compositions of studied melt inclusions.** The insets show the range of all melt inclusions. Pb isotope characteristics of olivine-hosted melt inclusions compared with modeling results (**a**). Locations of EM I and FOZO are from ref. 95. Fields for Pacific Ocean-type mantle, Indian Ocean-type mantle, EM I OIB, EM II OIB, and HIMU are from the GEOROC database (http://georoc.mpch-mainz.gwdg.de/georoc). Basaltic whole-rocks fields of NSP, CIP, MT (16–22°N), and AVP (except for Iwo Jima and Kita-Iwo Jima) are compiled from literatures[4,10,15,19,35–37]. The fields of Iwo Jima and Kita-Iwo Jima from southern Izu-Bonin arc with HIMU characteristics are from ref. 10,12,17. The field of Magellan seamount with HIMU characteristics is from ref. 71. Error bars with the 207Pb/206Pb and 208Pb/206Pb data are 2 standard errors (2SE). Modeling calculations indicate that mixing DMM with subducted sediments (SED) and AOC,

and then mixing with ~1.5% HIMU component could produce the Pb isotope compositions in NSP-24 group 2 melt inclusions. Modeling parameters can be found in the Supplementary Data 6. Primitive mantle-normalized spidergrams of trace element modeling results of NSP-24 calculated composite mantle component (**b**). The DMM is mixed with a bulk slab component comprising a mixture of 70% subducted sediment and 30% AOC, and then mixed with 1.5% HIMU lava to yield a composite component. This composite source is assumed to be contained in garnet-lherzolite consisting of 60% olivine, 20% orthopyroxene, 10% clinopyroxene, and 10% garnet[96]. About 1–3% partial melting of this composite component can produce trace element compositions that are similar to those of NSP-24 group 2 melt inclusions. The calculations use partition coefficients for silicate/basaltic melt from the GERM-KdD database (https://kdd.earthref.org/KdD/).

Partial melting model calculations attributed geochemical variations along the Mariana magmatic front to varying degrees of melting of a homogeneous mantle[43]. Geochemical studies of NMT lavas also show that the degree of mantle partial melting gradually increases northward as the rift propogated in that direction[14]. In contrast, Bloomer et al.[26] argued that melts of the Mariana arc magmatic front were produced by similar degrees of mantle partial melting. In this study, we estimate the partial melting degree of different groups of melt inclusions from ratios of incompatible trace elements (Supplementary Fig. 7). Although the results may vary as the simulation parameters change, the overall modeling trend shows that Mariana back-arc basin basalt is mainly derived from partial melting of spinel-bearing mantle peridotite, which is consistent with the results of ref. 43. In contrast, the Mariana arc magma source contains residual garnet, and the role of garnet increases northward (i.e., NSP-24 > NSP-21) (Supplementary Fig. 7). Different groups of NSP-21 melt inclusions show similar degrees of partial melting, whereas NSP-24 high-K melt inclusions appear to represent lower degrees of melting than do medium-K melt inclusions (Supplementary Fig. 7). However, the difference in melting degree is not enough to control observed incompatible element variations (such as K, Li, Rb, and Ba). Accordingly, the distinctive element compositions of these different groups of melt inclusions may reflect sources affected by variable subduction components rather than different degrees of partial melting of a homogeneous mantle source. This conclusion is also consistent with Pb isotopic variations.

## Subduction components in MT-18, NSP-21, and NSP-24 mantle sources

The source of most arc and back-arc magmas is the mantle wedge, which is affected by the addition of subduction components and exhibits enrichment of LILEs relative to HFSEs and REEs[1]. On the basis of trace element data, Peate and Pearce[40] proposed that the NSP subduction component is dominated by silicate melts derived from subducted pelagic sediments and AOC. NSP-21 and NSP-24 melt inclusions are enriched in LILEs and depleted in HFSEs, including negative Nb, Ta, and Ti anomalies (Fig. 2). These can be explained by residual or fractionating minerals containing these elements or by a

depleted mantle wedge. Some elements, owing to their differential compatibility in magma, can be used to further constrain the types of subduction components. We use incompatible trace element ratios, which change little during fractional crystallization and partial melting or during post-entrapment crystallization, to constrain the contributions of subduction components to the mantle source. For example, Th is less soluble than REEs and HFSEs in slab-derived fluids, whereas Ba is strongly partitioned into fluids[55,56]. In addition, Th and Ba abundances in subducted sediments are much higher than their abundances in the mantle[5,57].

As is illustrated in Fig. 4, NSP-21 and NSP-24 melt inclusions have highly variable Th/Yb and Ba/Yb ratios, suggesting that subducted sediment contributions are more variable than those of subducted AOC (Fig. 4). Although MT-18 melt inclusions have E-MORB-like trace element distribution patterns, they still show some influence of subduction components, such as negative Nb, Ta and positive Pb anomalies (Fig. 2), slightly elevated Th/Yb and Ba/Yb ratios (Fig. 4) and high Ba/Nb ratios (>7; Supplementary Data 3) as discussed by ref. 58. In addition, the Pb isotopic compositions of MT-18 melt inclusions covers almost the full range of 16-22°N Mariana Trough basalt compositions (Fig. 3a). This suggests that a wide range of subduction components and small-scale mantle heterogeneity interact beneath the Mariana Trough, which also explains the large variation of whole-rock geochemical compositions in the study area (Supplementary Figs. 2–4). Nevertheless, subduction components have less influence on the MT-18 magma source compared to the NSP magma source (Figs. 2, 4).

NSP-21 K-rich melt inclusions have especially high alkali contents and can be classified as alkaline series (Supplementary Fig. 2a), similar to the high-K melts of Kasuga 2 and 3 seamounts from the Kasuga Cross-Chain in the northern Mariana back-arc basin[2,49], suggesting that a high-K magmatic component is present beneath the volcanic front, as well as behind it in the Kasuga Cross-Chain (Fig. 1). In addition, NSP-21 K-rich melt inclusions have higher 208Pb/206Pb and 207Pb/206Pb ratios and Li, Rb, and Ba contents, but lower Cu, Zn, and Pb contents, than do coexisting NSP-21 medium-K melt inclusions (Figs. 2, 3, and Supplementary Fig. 4). Slab-derived fluids with high Li, Rb, and Ba contents can metasomatize the subarc mantle and contribute these elements to arc magmas. Potassium abundances in some arc lavas are consistently

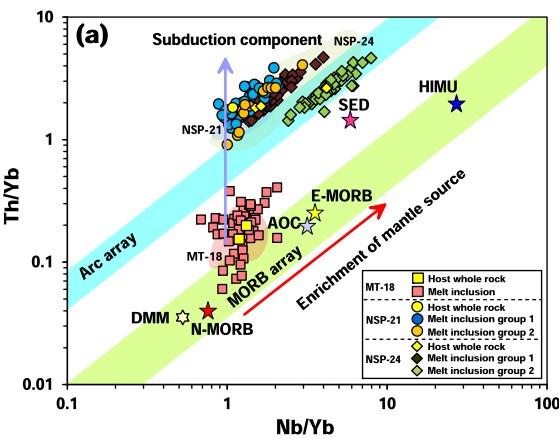

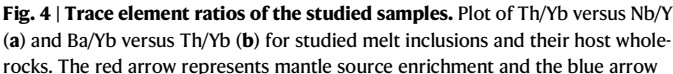

**Fig. 4 | Trace element ratios of the studied samples.** Plot of Th/Yb versus Nb/Yb (**a**) and Ba/Yb versus Th/Yb (**b**) for studied melt inclusions and their host whole-rocks. The red arrow represents mantle source enrichment and the blue arrow shows the addition of the subduction component. Subducted sediment is an important factor in changing the trace element ratios of samples. End-member data are from ref. 94 and Supplementary Data 6.

high[59,60], suggesting that a large amount of potassium is retained in subducted sediments by mica (biotite and/or phengite) to considerable depth, where it may be released by partial melting[61,62]. When sulfides (e.g., chalcopyrite, sphalerite, and galena) crystallize and/or separate from magma, Cu, Zn, and Pb partition into them[63], resulting in residual magmas that are highly depleted in chalcophile elements[64,65]. It should be noted that NSP-21 high-K melt inclusions have low MgO content (<3.07-3.58 wt.%) and a slight correlation between Pb isotopes and MgO (Supplementary Figs. 3, 6), indicating that these melts were products of fractional crystallization. Therefore, it is reasonable to speculate that fractional crystallization and/or sulphide immiscibility control the element anomalies coupled with subducted K-rich slab components that also affect the Pb isotopic compositions of the NSP-21 K-rich melt inclusions.

Except for a few elements (e.g., Rb, K, P), the two groups of NSP-24 melt inclusions have similar trace element distribution patterns, while their Pb isotopes differ significantly (Figs. 2, 3a). Medium-K melt inclusions in NSP-24 and NSP-21 have very similar trace element contents and Pb isotopic compositions, which are consistent with NSP whole-rock compositions (Figs. 2, 3a). In contrast, NSP-24 high-K melt inclusions show unusually low $^{207}Pb/^{206}Pb$ and $^{208}Pb/^{206}Pb$ ratios (note that the lowest $^{207}Pb/^{206}Pb$ and $^{208}Pb/^{206}Pb$ ratios correspond to the most radiogenic Pb), which is not reflected in NSP basaltic whole-rock compositions but is close to andesitic-dacitic lavas from Iwo Jima (24°45′N) and Kita Iwo Jima (25° 30′N) in the nearby VA[10,12,17] (Figs. 1, 3a). However, NSP-24 high-K melt inclusions are all basaltic and have different trace elements distribution patterns than Iwo Jima and Kita Iwo Jima lavas (Supplementary Figs. 2, 5), indicating different mantle source components and/or magmatic evolution processes. This suggests that medium-K melt inclusions represent partial melts of ambient NSP mantle, while the high-K melt inclusions come from different mantle source regions. Figure 3a shows that these low $^{207}Pb/^{206}Pb$ and $^{208}Pb/^{206}Pb$ ratios cannot be explained by mixing of depleted MORB mantle (DMM), subduction sediment, and AOC, but rather point to a HIMU end-member component. This is a key observation from our melt inclusion data.

**A HIMU-like component in the NSP magma source**
As one of the mantle end-members, the HIMU reservoir is defined by the composition of lavas from Mangaia and Tubuaii islands in the southwest Pacific Ocean and St. Helena in the south Atlantic Ocean, which have radiogenic Pb ratios ($^{206}Pb/^{204}Pb > 20.5$)[66,67]. To determine how the HIMU component was added to the magma source of NSP-24 high-K melt inclusions, we carried out a simulation calculation using a bulk components mixing model[68,69]. The results show that the injection

of less than 1.5% HIMU component into a composite of 99% depleted mantle and 1% mixture of subducted sediment and AOC (70:30) can explain the Pb isotopic compositions of NSP-24 high-K melt inclusions (Fig. 3a). To further test the possible petrogenetic link between the HIMU mantle component and the NSP magma source, we calculated trace element distribution patterns (Fig. 3b). Using results from the Pb isotope simulation (Fig. 3a), the calculated melt trace element patterns show that about 1–3% partial melting of this composite produces trace element patterns that are strikingly similar to those of NSP-24 melt inclusions (Fig. 3b).

The formation of the HIMU end-member with a high $^{206}Pb/^{204}Pb$ ratio requires long-term isolation (~ 2 to 3 Ga), and time-integrated enrichment of Th and U relative to Pb[70,71]. Two regions in the Earth where this can happen are the mantle transition zone (MTZ)[72,73] and the base of the lower mantle near the core-mantle boundary[74,75]. Geophysical evidence does not reveal any deep-rooted mantle plumes beneath the Mariana arc, which means that a direct contribution of HIMU from the deep mantle to the NSP magmatic source is unrealistic. In contrast, there are many ocean island basalt (OIB)-type seamounts with HIMU-like Pb isotopic characteristics outboard of the Mariana Trench (e.g., Wake and Magellan seamount chains; Figs. 1, 3)[76-79] which are subducted and provide more likely sources of the HIMU component (Fig. 3a). Therefore, we propose that the subduction of these HIMU seamounts is responsible for generating NSP enriched magmas (Fig. 5).

**Implications for subducted HIMU component recycling in the Mariana subduction zone**
For the IBM system, there are abundant seamounts in the western Pacific outboard of the Mariana arc (Fig. 1), many of which have HIMU characteristics and are subducting with the Pacific Plate[76,80,81]. In addition, the HIMU-type seamount subduction signal has also been recognized in the central Mariana forearc metabasalts (16–18°N)[8,82]. Pb isotopic compositions of NSP-21 and NSP-24 medium-K melt inclusions are close to those of the FOZO end-member (Fig. 3a). FOZO (FOcal Zone[83]) is usually interpreted to have formed in a manner similar to that of HIMU, but its overall formation age is younger than that of HIMU[79,84]. As shown in Fig. 3a, the addition of a small amount of HIMU components (~ 0.5%) to the mantle source can explain the Pb isotopic variation of NSP-24 and NSP-21 medium-K melt inclusions and most CIP, NSP, and AVP whole-rock compositions. Therefore, we speculate that a slight HIMU-like component exists widely in the Mariana subduction system, as previously suggested[10,19,85].

In contrast to arc magmas such as NSP-21 melt, obvious HIMU characteristics only appear in NSP-24 melt inclusions associated with the earliest stage of rifting (Fig. 1). This suggests that the HIMU

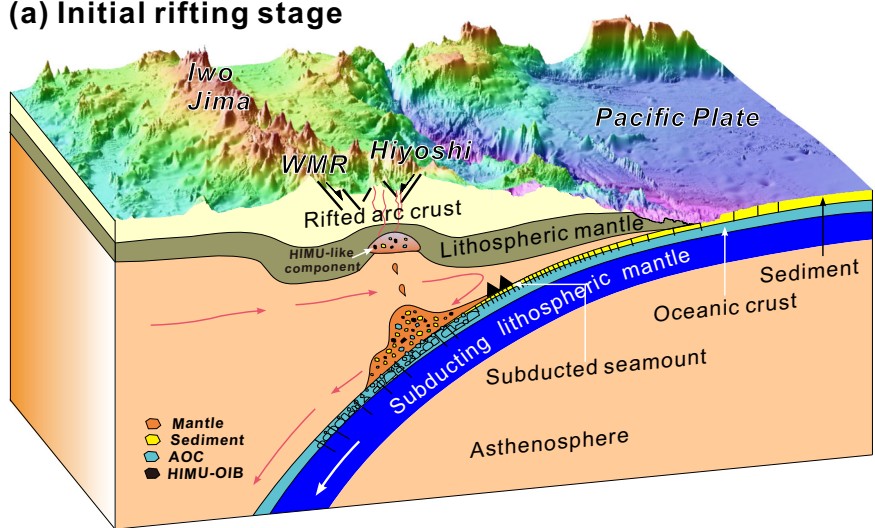

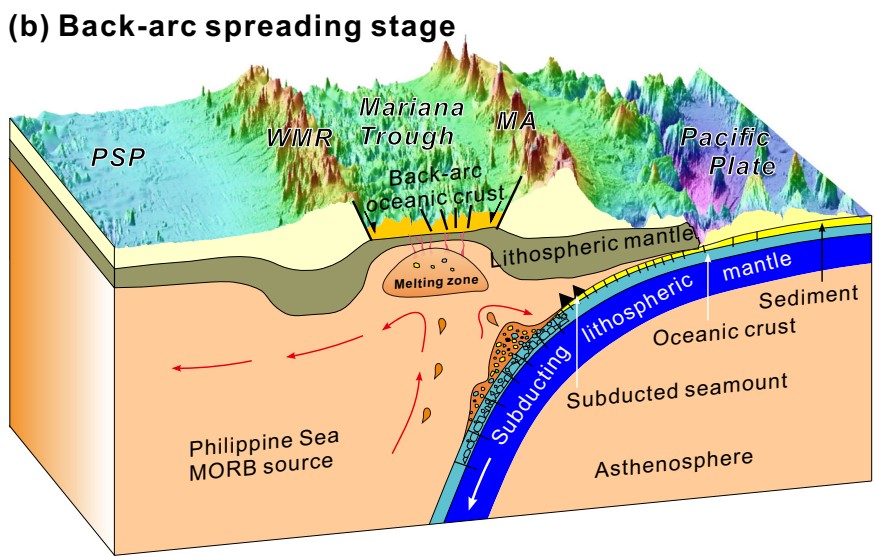

**Fig. 5 | Conceptual model for the recycling of subduction components in the Mariana subduction zone.** The base map data is from SRTM15 + V2.5.5[93]. Subducted seamounts together with associated volcanogenic sediments, AOC, and mantle form a HIMU-like component, which is diapiric and stored in the lithospheric mantle. The HIMU-bearing component is remobilized and undergoes a small degree of decompression melting when rifting happens with the crust thinned and melt in this initial stage of the rift has significant HIMU characteristics (**a**); Back-arc basin spreading center is located far away from the trench and is mainly fed by a large amount of MORB-like magma, no HIMU component is injected or this component is consumed or swamped with a large partial melting degree during the back-arc spreading stage (**b**). WMR – West Mariana Ridge; MA – Mariana Arc; PSP – Philippine Sea Plate.

component is only mobilized into arc magma by rifting. HIMU components from subducted seamounts may migrate and be stored in the lithospheric mantle. Rifting thins, heats and decompresses the lithosphere, remobilizing some components in it, including rifting HIMU-bearing components (Fig. 5a). This interpretation is consistent with the high Nb/Ta ratios in NSP[25] and the anomalous low-velocity zone in the upper mantle beneath the region of rift propagation[86]. As rifting progresses, asthenospheric magma sources dominate over lithospheric sources, more MORB-type components are involved with the HIMU component gradually becoming less important. Once spreading begins, the HIMU component is swamped by upwelling asthenospheric MORB-type magma and the HIMU signature disappears (Fig. 5b). The element and isotope compositions of MT-18 melt inclusions show MORB-like characteristics (Figs. 2, 3). This may be because the back-arc basin spreading center is located far away from the trench, and is mainly fed by a large amount of MORB-like magma. If so, then no lithospheric HIMU component is present, or it has been diluted. Our melt inclusion study demonstrates that early rift magma retains more complete information about subducted components and that a significant HIMU-like component in the Mariana subduction system may be more widespread than previously thought, at least to the north of 24°N, which may be the key to producing arc potassic/shoshonitic magmas.

## Methods

Selection and preparation of melt inclusions were carried out at the Guangzhou Institute of Geochemistry, Chinese Academy of Sciences. Melt inclusions were re-homogenized and quenched using a high-temperature furnace following procedures in ref. 87. A platinum packet containing olivine crystals was heated at 1250 °C for 10 min to melt the crystal-glass mixture. Then, the packet was quickly raised to the top of the furnace tube to achieve quenching. Melt inclusion-bearing olivines were selected, mounted on epoxy resin disks, and polished until the melt inclusions were exposed for analysis. The major

and trace element analysis of host whole-rocks are presented in the Supplementary Information.

## Olivine and melt inclusion major element analysis

Major elemental compositions of melt inclusions and host olivine phenocrysts were analyzed with a JEOL JXA-8230 electron probe microanalyzer (EPMA) at the Key Laboratory of Submarine Geosciences and Prospecting Techniques, Ministry of Education, Ocean University of China, Qingdao. The operating conditions were 15 kV accelerating voltage, 20 nA beam current, and 2 μm beam diameter for melt inclusions and host olivines. Na and K were measured first with 10 s for the peak and 5 s for the background to minimize volatilization loss. Analytical precisions for major and trace elements were better than 1% and 5%, respectively.

## Melt inclusion trace element analysis

After EPMA analysis of melt inclusions, their trace elements were measured using inductively coupled plasma mass spectrometry (ICP-MS) equipped with a 193-nm (ArF) laser ablation system in the Institute of Oceanology, Chinese Academy of Sciences. The carbon coating was completely removed from each epoxy resin disk before laser ablation. Laser conditions were set to 25 μm beam size, ~4 J cm$^{-2}$ energy density, and 5 Hz repetition rate. Each analysis consisted of a 20 s gas blank with the laser off and a 20 s sample ablation signal with the laser on. Every eight sample analyses were followed by an analysis of the external reference glasses of BCR-2G, BHVO-2G, and BIR-1G to correct for sensitivity drift and mass discrimination. Si determined by EPMA was used as the internal standard element for correction. All analyzed data were processed using the offline software ICPMSDataCal 11.8[88]. Results show that the measured values of reference standards are in good agreement with the recommended values (Supplementary Data 4). The accuracy and precision of most trace element analyses were better than 10% and 20%, respectively (Supplementary Data 4).

## Melt inclusion Pb isotope analysis

After EPMA and LA-ICP-MS measurements, melt inclusions with diameters >40 μm were chosen for in situ Pb isotope analysis using a Neptune Plus multiple collector inductively coupled plasma mass spectrometer (MC-ICP-MS, Thermo Fisher Scientific, Bremen, Germany) coupled to a 193 nm (ArF) COMPexPro 102 laser ablation system in the Guangxi Key Laboratory of Hidden Metallic Ore Deposits Exploration, Guilin University of Technology, Guilin, China. Prior to this procedure, each disk was washed with purified 0.1% nitric acid and ultrasonically cleaned. Detailed procedures for Pb isotope analysis followed those of ref. 89. The laser parameters were set with a repetition rate of 6 Hz, laser beam diameter of 32 μm, and energy of 5 J cm$^{-2}$. Helium was the carrier gas (650 ml min$^{-1}$). Before samples were lasered, 30 s were used to detect the gas blank, and then 30 s were used to laser ablate the samples and collect the ablation signals. In order to improve instrumental sensitivity, an X-skimmer cone and a Jet sample cone were used. All isotopic signals were detected with ion counters under static mode. The NKT-1G international basaltic glass standard[90] was chosen as the external standard. Basaltic glass standards NIST 614 and BHVO-2G were analyzed before and after each batch of eight melt inclusion analyses to monitor instrumental drift. Here, we present the ratios of $^{20x}Pb/^{206}Pb$ (x = 208, 207) and $^{20y}Pb/^{204}Pb$ (y = 208, 207, 206) of the melt inclusions. Data processing followed ref. 91. Instrumental mass bias was corrected using the standard–sample–standard bracketing method and any outliers (>2 SD) were excluded. The Pb isotope measurement of every unknown sample was carried out before and after the measurement of standard glass for mass bias correction. Analyses of NIST 614 and BHVO-2G during this study yielded averaged $^{20y}Pb/^{204}Pb$ and $^{20x}Pb/^{206}Pb$ consistent with the recommended values. For most analyses, precisions were better than 0.39% for $^{20x}Pb/^{206}Pb$ and 1.22% for

$^{20y}Pb/^{204}Pb$ (2RSD, $n = 20$), whereas accuracies were better than 0.25% for $^{20x}Pb/^{206}Pb$ and 0.28% for $^{20y}Pb/^{204}Pb$ (Supplementary Data 5). Due to the small melt inclusion lasered size (32μm) and low relative abundance of $^{204}Pb$, precisions of $^{20y}Pb/^{204}Pb$ are lower than the $^{20x}Pb/^{206}Pb$ (Supplementary Data 5). Therefore, we emphasize ratios of $^{20x}Pb/^{206}Pb$ for melt inclusions in this study.

## Data availability

The authors declare that the data generated or analyzed during this study are available within its supplementary data in a Source Data file deposited at EarthChem Library (https://doi.org/10.60520/IEDA/113107)[92]. Source data are provided with this paper.

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

## Acknowledgements

We thank Le Zhang and Xiaohong Wang for assisting with the melt inclusion preparation and trace element analysis, and thank to Zhigang Zeng for providing part of the samples in our preliminary research, which provided us with inspiration for this study. Funding for this research was provided by the National Natural Science Foundation of China (NSFC) (42006052) to X.H.L., NSFC (42121005, 42488201) and Scientific and Technological Innovation Project of Laoshan Laboratory (LSKJ202204401) to S.Z.L., the Fundamental Research Funds for Central Universities (2023000-842172002、842172003) to X.H.L., and NSFC (42072061) and the Shandong Excellent Young Scientist Grant (ZR2022YQ32) to L.C. This is UTD Geosciences contribution #1714.

## Author contributions

X.H.L. and S.Z.L. conceived and designed the study. X.H.L., O.I., R.J.S., S.Z.L., I.S. and Y.H.S. wrote the manuscript. X.H.L., Z.Q.L. and H.X.Y. analyzed the data. X.H.L., O.I., R.J.S., S.Z.L. and L.C. interpreted the data. All authors contributed to discussions.

## Competing interests

The authors declare no competing interests.
