## [Peer Review File · Nature Communications]

REVIEWER COMMENTS

Reviewer #1 (Remarks to the Author):

This is an interesting and well-written paper with a LOT of new major and trace element and Pb isotope data for olivine-hosted melt inclusions (OHMI) and their host rocks from three locations in the Mariana arc and backarc. The new data extend the geographic range of previously known things, and include something altogether new: 12 weird andesitic OHMI (7% of those analyzed) that have unusually low Pb contents and EM1-like Pb isotope ratios in a basalt from a seamount in the northern volcanic front of the arc. The data seem reliable and warrant publication, and much of the interpretation is reasonable, but several fundamental topics need attention during revision.

First, the paper presents itself as a study of geochemical change during the transition from rifting to spreading in a backarc basin. However, this study cannot separate the along-strike spatial variations in the Mariana system (i.e., what is being subducted and the depth of the slab beneath eruption sites) from the effects of the rifting-to-spreading process. Second, the OHMI are less novel than claimed. Third, the paper over-states the importance, and over-interprets the origin, of the weird OHMI at NSP21. I will elaborate on those three points below.

An alternative emphasis of the paper, which is there already but under-played, might be how using OHMI can discover extreme melt mixing components that are diluted in erupted magmas in three different tectonic settings in the Mariana system. The second half of the paper is mostly about this topic; it's only the title and introduction that lead readers (at least this reviewer) to expect the paper to be about how and why magma changes from backarc rifting to spreading. Revision could be mostly about organization and emphasis.

The first fundamental topic is how to separate the spatial effect from the stage of backarc evolution. Latitudinal variations in the Marianas have been known for many decades and attributed by some of the co-authors of this paper, and others, to subduction of enriched seamount chains in the vicinity of Iwo-To (aka Iwo-Jima). That enriched signal decreases southward along both the volcanic front and the Mariana Trough (i.e., from 25° to 21° and beyond) and northward to 26°. The authors need to explain why their observations cannot be explained just by differences in what is subducted at those latitudes, or how much is added to the mantle at various distances behind the volcanic front or above the slab. That is, the slab component may have always differed in kind or amount or ubiquity between at 24°, 21° and 18°N.

The rift-to-spread argument also begs the question of when and how a HIMU-type component got as far south as 18°N beneath the MT where there is no slab currently. Did appropriate seamounts get subducted there too when the slab dip was shallower? Or did the slab-contaminated mantle flow south as the arc unzipped? Or was the HIMU character inherited from the Oki-Daito plume under the Philippine Sea Plate (Ishizuka et al., 2022) and is unrelated to subduction? Answering this may require careful comparison of the enriched components in the Mariana Trough and arc in light of recent data.

The NSP24 host rocks and their OHMI are very similar to the Hiyoshis with the principal difference being their lower Ba/La, Pb/Ce, and Sr/Nd ratios despite higher La/Yb. Those differences may reflect their

reararc setting, such as in the Kasuga and Guguan cross-chain (Stern et al., 1993, 2006). All three samples are from the eastern scarp of the remnant arc, ~40 km behind the volcanic front. They are 2.5-3.3 Ma and may pre-date rifting. At best, they are from the earliest stage of extension. I agree with the authors that their mantle source probably underlay the arc just before, or as, it started to rift, but these are samples of the reararc 3 m.y. ago, not the rift now.

In light of this, I suggest that the paper be re-structured to focus more on the differences in slab components and less on the rift-to-spread process. If the authors retain an emphasis on the latter, then they must address other places where that process can be, and has been, studied more thoroughly than here. These are primarily in the southern hemisphere including the Manus, North Fiji, and Lau Basins, and Kermadec Trough. Spreading at the East Lau and Futuna Spreading Centers in the Lau Basin reaches as close to the volcanic front as the WMR is to Hiyoshi. If anything, the nature of the slab component and its effects on mantle melting during initial rifting can be studied there even better than in this paper (e.g., Bezos et al., 2009; Escrig et al., 2012). The rifting stage is known in the greatest detail in the Havre Trough (e.g., Gill et al., 2021). The current paper is provincial in this respect.

Second, the OHMI are less novel than claimed. The title proclaims that a HIMU-like has been revealed by their melt inclusions, but it was discovered almost 60 years ago by Tatsumoto at Io-To, then called Iwo Jima, in one of the first papers about Pb isotopes in volcanic rocks! The Pb isotopes in the Group 2 NSP24 OHMI are similar to those in Io-To (Tatsumoto, 1966; Sun and Stern, 2001) and Kita-Iwojima (Ishizuka et al., 2010) even though the major and trace elements differ. The opposite is true for the Group 2 NSP21 OHMI that are similar to the high-K basalts of the nearby Kasuga 2 and 3 seamounts in everything except their weird Pb isotopes and low chalcophile element contents.

Third, the paper over-states the importance, and over-interprets the origin, of the weird Group 2 OHMI at NSP21 that are a new discovery in this paper. By “weird” I mean their low chalcophile element contents and their Pb isotope ratios. The host rock (T1-3) is similar to the high-K basalts at the nearby Kasuga 2 and 3 seamounts (Fryer et al, 1997). The OHMI are similar in MgO and some other oxides to the dacites from Kasuga 3. It seems like special pleading to infer an entirely different slab source for these 3 wt% MgO OHMI in Fo70 olivine when so much is similar to nearby rocks. Why not appeal to something that just affects chalcophile elements during differentiation? For example, sulfides are known to fractionate Pb isotopes such that coexisting pyrite-sphalerites can have at least the range of Pb isotopes as in these OHMI (Yuan et al., 2015: Science China: Earth Sciences). After all, the Sr-Nd-Pb-Mo isotopes all vary considerably as a function of MgO at this location (Li et al., 2021), so it is an open system. Adding an exotic hydrothermal fluid to differentiated, Pb-poor melt might be an easier explanation than the 80 lines of speculation that start at l.331.

Moreover, whatever their origin, they are a tiny fraction of the OHMI that were studied, and they can be only a small mass fraction of magma at the volcanic front including their host rock. Try mass-balancing the OHMI to equal the host rock. The claim that this component “dominates the Mariana subarc mantle” (l. 542) is a vast over-statement. These are an intriguing discovery on the flanks of one seamount, but there is no evidence that they are a common component of Mariana volcanic front lavas.

An alternative emphasis for the paper might be how OHMI preserve slab component end-members at

each site that are diluted in whole rocks (i.e., the subduction components section in I.289ff). To me, their most striking discovery is one that they do not discuss even though it is the best evidence for their claim that a wide range of slab components, including volcanoclastic as well as pelagic sediments, might be present beneath the Mariana Trough as far south as 18°N. The Pb in their OHMI from 18°N covers the full range of Pb isotopes of the entire Mariana Trough from 16-23°N, extending from the ambient mantle of Woodhead et al. (2005) to their most enriched MT basalts. There is even a negative correlation between $^{208}\text{Pb}/^{206}\text{Pb}$ ratios and Ba and Th-enrichment in both the OHMI and whole rocks. For NSP24, the OHMI data mean that they found the extreme HIMU component as far south of Io-To as Ishizuka et al. (2007) found it to the north at Kita-Iwojima, and that it occurs even in the reararc where Pb/Ce ratios are lower, K₂O is higher, and the slab component is melt-like. This may be important but will require careful consideration of published data for all samples from 23°-26°N. As the authors already say, the NSP21 OHMI data mean that small amounts of a high-K component are present even at the volcanic front as well as behind it in the adjacent Kasuga cross-chain.

This paper's discussion of the nature and origin of slab components is most persuasive about the HIMU component in I.411ff, and that topic is in the paper's title. The discussion seems reasonable to me, but I do not think that the data in this paper add new insight into the general origin of HIMU. It's not like Zhang et al.'s (2022) discovery of high $\delta^{66}\text{Zn}$ values in HIMU lavas from the Cook-Austral islands and St. Helena. Consequently, while I found nothing wrong in that section, I do not think that it warrants such length. The overall model for NSP24 melts is essentially the same as in Ishizuka et al. (2010).

In addition to those main points, here are some more specific things to address during revision.

The paper needs much greater clarity about exactly what samples were studied.

- MT18. Are the analyses of L1 and L3 in Supp.Table2 new analyses of the samples by that name in Zhao et al. (2016) that in turn came from Tian et al. (2005)? If so, then by what analytical methods? The data differ between this paper and Zhao/Tian by almost 2 wt% MgO and 25 relative% Na₂O, for example, and by a factor of 4 in some trace elements. Or, are these new analyses of the same or similar samples from 18°12'N that have a different name (T2-1,2,3) in another recent paper about Mo isotopes by the same first author of this paper (Li et al., 2021:Geology)? Give the lat/long and water depth at which the samples in this study were collected, details of their collection, and how they compare to other samples from the site. Then discuss how these two samples relate to the heterogeneity known along the MT between 18-19°N (segment 6 of Pearce et al., 2005). Trace element and Pb isotope ratios can vary significantly in <1 km in this segment, and all the samples in the Li and Zhao/Tian papers plus those from the same area in Chen et al. (2021) are more enriched than many (cf. Pearce et al., 2005; Woodhead et al., 2015). Neither sample in this paper meets the criterion ($\text{Ba}/\text{Nb}<7$) for unmodified (ambient) mantle wedge of Woodhead et al. (2015) whose Pb isotopes plot about where the letters CMT are in Fig. 3a of this paper. That strengthens the authors' inference that even their MT18 samples have a significant slab component but they chose an especially enriched example to study. They also might note that everything north of 19.8°N -- where the seafloor fabric deteriorates at the north end of Pearce et al.'s (2005) segment 3, west of Asuncion and Maug at the volcanic front -- appears enriched in a slab component based on the recent results of Chen et al. (2021) using old SIO dredges. Whether that change reflects melting out of a uniform slab component during rifting-to-spreading, or a change in the amount or kind or ubiquity of a slab component, is the question.

•NSP21. In reply to questioning, the authors said that sample T1 in SuppTable2 is a new analysis of a sample like T1-3 in Li et al. (2021), and they provided a map showing its location between Eifuku and Kasuga on the volcanic front. That reference and map need to be in the paper. Also, it should be noted that Eifuku has the lowest Nd and Hf isotopes south of the NSP, consistent with the addition of the most sediment (mixed pelagic-volcaniclastic) to its mantle source, according to Woodhead et al. (2015). The sample from this location with the lowest Nd isotope ratios in Li et al. (2021) also has the lightest Mo, consistent with that information.

•NSP24. In reply to questioning, the authors said these samples are from the West Mariana Ridge (Ishizuka et al., 2010) including D13 that was not discussed in that paper. Again, this information and map need to be in the paper, and the map should identify the Hiyoshis.

I.59. Few if any MT basalts cannot be distinguished from MORB using criteria like H₂O/Ce or Ba/Nb. Even Th/Yb vs Nb/Yb (Fig. 4a) distinguishes most.

I.68. What does “typical” mean? That requires comparison to other examples like those in the southern hemisphere. This is one example of un-necessary over-reach because of emphasizing the process of backarc basin evolution.

I.79. Why is the N-NSP “ideal”? As noted above, the chosen samples may even pre-date rifting.

I. 173. Why “arc” tholeiites?

I.186. The OHMI are NOT more enriched in K₂O/TiO₂ than E-MORB, but they are enriched in Ba/Nb when MgO <7 wt%. The latter is how Woodhead et al. (2015) distinguished their source as having a slab component. Most also are enriched in Th relative to ambient mantle, although it is important to compare them to “Indian”-type MORB.

I. 261ff. Consider breaking this long paragraph into two, with the first being about fractional crystallization, and the second starting at I.266 about partial melting. For NSP24, the latter depends on Supp.Fig.6 that in turn depends on the oddly low HREE in all the NSP24 OHMI that is rare in arc rocks. That accounts for the high Sm/Yb despite low La/Sm in these OHMI. The authors attribute that atypicality to 20% residual garnet in a depleted peridotite mantle source. That much garnet is unlikely even for fertile DM. Moreover, it seems impossible to mass balance the heavy REE in these OHMI relative to their host or other nearby rocks. Therefore, the model in Supp.Fig.6 is unconvincing. If not an analytical artifact, then might such low HREE result from adding eclogite melt from the slab to the reararc mantle?

I.288. By “heterogeneous sources” do you mean a variable addition of slab component to a homogeneous ambient mantle? That seems like the simplest, and most conventional, option.

I.293. Why not say silicate melts instead of silicic fluids?

I.302. Why not say that Th is less soluble than REE and HFSE in “fluid”, but then cite experimental evidence rather than inferences from other field studies.

I.312. Wouldn't adding the same mass of sediment melt to variably depleted mantle have the same effect?

I.315-318. I haven't chased down all these references, but I didn't think that the first three appealed to a sediment melt component in MTB. The classic Stolper and Newman (1994) paper was about a low-density fluid, which this paper's authors seem to prefer too.

I.340. What differs from the high-K basalts at Kasuga 2 and 3? The Pb isotope ratios are higher and chalcophile element contents are lower, but what about everything else?

I.411. NSP21 and 24 rocks have quite different Nd and Hf isotope ratios so they cannot have similar sources even if their Pb is similar.

I.420. The Pb isotopes of the high-K OHMI aren't then same as in the Hiyoshis or WMR, but they are like Iwo-Jima, which requires a de-coupling of Pb from K between those sites, which is cool.

I.478. This model for the HIMU component is essentially the same as in Ishizuka et al. (2010) and seems reasonable. However, please also apply it to the Group 1 OHMI in the same sample. Presumably they have less HIMU (carbonatite?), but why then do they have lower Nb/Yb and higher Ba/La and Th/Nb than Group 2?

Tables:

- The uncorrected K₂O and P₂O₅ entries for Type 2 NSP21 inclusions are transposed in Supp.Table1.
- I strongly urge the authors to obtain and use IGSN numbers for their samples prior to publication. Otherwise, as more data are published for these and other samples, it will become increasingly difficult to keep track of what is being studied using just a D- or T-number.
- Is there a rationale for the sequence of sample numbers in Supp.Table1?
 - > Specify the host rock of the melt inclusions in MT18 and NSP24, and group the OHMI in each by host rock.
 - > If any of the OHMI are from the same olivine, then group them together too.
 - > Beyond those two principles, why not present the data in order of decreasing Mg# of the host or the melt inclusion? That would help readers to see the data in a geochemical context.

Figures:

- Figure 1 or a Supplementary equivalent needs to have much more detailed maps showing the location of the samples; something like Fig. 1c of Ishizuka et al. (2010) for each site.
- Figure 2. Perhaps as Supplemental equivalents, I wanted to see a least-squares best fit to mass balance between OHMI and host rock for c+d, and how the OHMI compared to whole rock compositions for the Hiyoshis and Iwo-To (Iwo-Jima) for e+f.
- Figure 3a. This is such a small scale and has so much white space that it is hard to see details. Consider reducing the x-axis to 0.80-0.88 and then showing HIMU and EM1 in an inset because the data are more important than the model. I assume that the dashed yellow line is for NSP, but is there also a buried dashed red line for CIP, and is the white dashed line for CMT? The AVP, Iwo-Jima, and Kita-Iwojima, need

to be added. All this should be explained in the caption. Why is the MORB source placed outside even the Pacific field? Why not place it inside the Indian field since the MT ambient mantle is Indian-type?

Reviewer #2 (Remarks to the Author):

review of Li et al: A HIMU-like component in the magma source during initial arc rifting revealed by melt inclusions

This manuscript presents new major and trace element concentrations and Pb isotope compositions for melt inclusions from lavas collected in three different regions of the Mariana arc: the mature back-arc, the northern portion of the main arc front, and the northern edge of the back-arc (NSP-24) where rifting is taking place. The authors show that all three locations exhibit distinct chemical and Pb isotopic compositions, largely confirming previous bulk rock studies from the same regions. The main difference to previous bulk rock work is that a group of melt inclusions from the NSP-24 region exhibit anomalous Pb isotope ratios that bear some similarities to so-called HIMU ocean island basalt lavas. After considering different scenarios of the HIMU origin, the authors conclude that it most likely represents seamounts that were subducted together with the Pacific plate. As such, no HIMU mantle source is present in the sub-arc Mariana mantle wedge, it is simply carried by the incoming subducted slab.

The quality of the data is good and, overall, I agree with the authors that their general interpretation is the most likely to account for a HIMU source in subduction-related lavas. In general, the paper certainly is worthy of eventual publication, but I am less sure of just how novel the paper really is given that the HIMU story feels more like an addendum to the generally well-studied Mariana arc system. I suppose it is up to the editor to determine if that merits publication in Nature Communications.

One major problem I had while reading this paper was that all the figures were presented in low resolution with very small font sizes. I, therefore, really struggled to see some of the details of the plots. I would urge the authors to fix this and recommend the Nature Communications do not allow papers with almost illegible figures to go out for review.

When diving into a more detailed view of the manuscript as written I have some issues, both with the way the interpretations are presented as well as the processes that are implied to drive the subduction processes. I outline these in more detail below:

1. The paper is far too long and some of the interpretations, while I generally agree with them, are argued in a convoluted way. For example, the section on the origin of the NSP-21 K rich lavas is almost four pages long. Yet, the authors end up concluding that the origin of these lavas is mostly due to assimilation-fractional-crystallization (AFC) type processes. I think this makes sense, but the authors could likely have stated this in less than a page. Instead of providing a long-winded discussion of other processes, I don't understand why they didn't just go straight to the strong evidence for their conclusion (low chalcophiles, low MgO, correlations between Pb isotopes and MgO, and more). Supplementary figures 3, 4, 5 do a great job of documenting all this.

Another example is the very long discussion the authors present on the origin of HIMU mantle sources.

This includes a detailed account of arguments supporting carbonate involvement in HIMU generation. However, since the main conclusion of the paper is that the HIMU signature comes from a seamount, I do not see any reason to even mention carbonates or carbonatites. The only evidence the authors present that they claim support a carbonatite involvement in the HIMU source is figure 5, which I find very tenuous. While the trace element ratios shown there are distinct from the other melt inclusion groups, they are only very slightly offset towards a carbonatite source and what they fail to mention is that the observed values fully overlap with normal mantle values. Hence, there is nothing particularly 'carbonatite-like' about the observed trace element data. In the end, none of that matters for the interpretation of the origin of the HIMU signature in the NSP-24 samples. It makes sense that it is a subducted seamount and whether that seamount had a carbonatite or an altered oceanic crust influenced mantle source is irrelevant.

2. There are some inconsistencies between the processes the authors seem to infer occurring during subduction and what has been proposed in the literature. For example, the authors claim that the mantle wedge is metasomatized by subducted sediments. To me it looks like these sediments are simply just bulk sediments. But typically, studies have suggested that sediments will experience partial melting before entering the mantle wedge and resulting in metasomatism. If the authors want to pursue such an interpretation, they would need to use sediment melts calculated based on one of the available experimental studies that have investigated such a process (eg Hermann and Rubatto, 2009). Including this calculation will primarily affect the trace element ratios of the resulting mantle source as well as the proportions of sediment required to explain the Pb isotopes.

Furthermore, I find it difficult to reconcile the HIMU seamount as a bulk component in the mixing calculations because that would require high degrees of partial melting of what is essentially basaltic oceanic crust. I am very doubtful about such a partial melting process since oceanic crust becomes eclogitic at sub-arc depths. In turn, melting of eclogite only happens at very elevated temperatures and also results in very specific geochemical signatures, such as high Sr/Y ratios (eg adakites). As far as I can tell, this is not observed in the studied samples.

If the authors prefer to consider bulk sediment and oceanic crust mixing processes for their Pb isotope and trace element calculations, then they are basically inferring so-called melange formation (eg Marschall and Schumacher 2012; Nielsen and Marschall 2017). Instead of the long-winded discussions of the NSP-21 samples and the origin of HIMU, it might make more sense to present a discussion of different subduction processes like mantle metasomatism and melange formation. To me that would be more directly related to the presented data set and would also, in my opinion, make the paper of broader more general interest that might be better suited to Nature Communications.

I also have some more specific line-by-line comments that the authors should address prior to resubmitting a revised version:

Line 108: I looks like the latitudes are wrong for the SMT

Line 155: what about re-equilibration? You just mention it here but do not further explore it. This process is the only one that is likely to affect isotopes and incompatible trace element ratios, so that

seems more important to discuss than the PEC

Line 194: delete significance

Line 306-308: you should note that some sediments can have very large Ba enrichments with Ba/La and Ba/Th ratios that exceed those of arc lavas. Hence, without knowing the Ba contents of subducting sediments outboard of your arc lavas, you cannot necessarily attribute Ba enrichments to fluids (neither from oceanic crust nor from sediment). As shown in recent Ba isotope papers on arc lavas, there is little evidence to suggest that Ba in arc lavas is primarily supplied by ocean crust derived fluids.

Line 310: I don't see this. Ba/La is quite variable, whereas Ba/Th is more constant. This may imply that Ba and Th are both coming from sediment and that fluids are less important in supplying Ba to the mantle source.

Line 315-317: I do not follow what evidence you have presented that supports this conclusion about the slab component in the mature back-arc. The subduction component is generally very small and, therefore, is difficult to determine the origin of.

Line 436-439: I don't think the paper by Zhang et al makes a strong case for carbonatite origin of HIMU. Zn isotopes are essentially unexplored in subduction zones and the paper presents no information about what might happen to Zn isotopes in a residual slab. So to me it is equally possible that small amounts of Zn isotope fractionation occurs during slab dehydration. I also cannot recognize the description of the Zn isotope values as 'exceptionally high' since they are only about 0.1 permil heavier than MORB and even less compared with other OIBs.

Line 441-443: I don't really agree that this is clear. The NSP-24 samples are only slightly plotting towards the carbonatite field and I don't see a clear reason for why carbonatite needs to be implicated.

Line 436-455: I do not understand the reason for the authors wanting to push the argument of HIMU being a carbonatite source. I don't think it has much to do with their data set.

Line 477: why use a generic marine sediment field instead of sediments outboard of the Mariana arc?

Line 478-481: This text seems to imply bulk mixing processes were considered. As I note above, such a process is not compatible with mantle metasomatism but instead more reminiscent of melange formation followed by partial melting.

Line 492-494: I still see no particular evidence to support that the HIMU component in the NSP has anything to do with carbonatite or carbonated mantle.

Line 509-527: I think the authors would be better served by deleting this whole paragraph.

Line 546-554: Alternatively, it is more of a random chance problem. There will not be HIMU seamounts subducting all the way along the arc. Maybe there is a HIMU seamount subducting under the NSP-24,

but not under NSP-21? Also, the mature back arc samples only carry a very minor slab component, which is very difficult to identify the exact nature of because it only causes a small modification away from the MORB field. So that, in itself, might not have anything to do with a progressive depletion of the mantle wedge. In any case, the NSP-24 samples only reveal a quite dilute HIMU signature, which, when further strongly diluted, would be essentially impossible to distinguish from any other slab component.

Line 556: I don't see how this reveals it is more common. It basically requires subduction of seamounts with a very specific HIMU composition. Whether these types of seamounts are more common than we think is another question. If yes, then it is possible that more arcs could include such a component.

Line 634: the authors need to describe how they corrected or instrumental mass bias. And how variations in mass bias were accounted for.

Line 634-641: Was the laser spot size the same for samples and reference materials?

Based on the SE of individual melt inclusions and the reference materials, it looks like they must have been similar. However, there is a large number of melt inclusion measurements that have significantly higher SE counting statistics than the reference materials. So these likely have greater error bars than the 0.39% indicated for the 208/206 and 207/206 ratios and 1.22% indicated for ratios with 204 involved. I think the authors need to consider if some of their Pb isotope variations (especially within a specific group) might be due to analytical scatter. This is not the case for the variation between groups which is much larger than what can be attributed to analytical scatter.

Response to reviewer's comments:

Reviewer #1 (Remarks to the Author):

This is an interesting and well-written paper with a LOT of new major and trace element and Pb isotope data for olivine-hosted melt inclusions (OHMI) and their host rocks from three locations in the Mariana arc and backarc. The new data extend the geographic range of previously known things, and include something altogether new: 12 weird andesitic OHMI (7% of those analyzed) that have unusually low Pb contents and EM1-like Pb isotope ratios in a basalt from a seamount in the northern volcanic front of the arc. The data seem reliable and warrant publication, and much of the interpretation is reasonable, but several fundamental topics need attention during revision.

First, the paper presents itself as a study of geochemical change during the transition from rifting to spreading in a backarc basin. However, this study cannot separate the along-strike spatial variations in the Mariana system (i.e., what is being subducted and the depth of the slab beneath eruption sites) from the effects of the rifting-to-spreading process. Second, the OHMI are less novel than claimed. Third, the paper over-states the importance, and over-interprets the origin, of the weird OHMI at NSP21. I will elaborate on those three points below.

An alternative emphasis of the paper, which is there already but under-played, might be how using OHMI can discover extreme melt mixing components that are diluted in erupted magmas in three different tectonic settings in the Mariana system. The second half of the paper is mostly about this topic; it's only the title and introduction that lead readers (at least this reviewer) to expect the paper to be about how and why magma changes from backarc rifting to spreading. Revision could be mostly about organization and emphasis.

Response: We are very grateful for your professional and constructive comments on our manuscript, which are very helpful to improve the quality of our manuscripts. We have taken some of your suggestions and made substantial revisions to the manuscript. We have changed the title to "A HIMU-like component in Mariana Convergent Margin magma sources during initial arc rifting revealed by melt inclusions". We hope you will be satisfied with our response and efforts in the revision process.

The first fundamental topic is how to separate the spatial effect from the stage of backarc evolution. Latitudinal variations in the Marianas have been known for many decades and attributed by some of the co-authors of this paper, and others, to subduction of enriched seamount chains in the vicinity of Io-To (aka Iwo-Jima). That enriched signal decreases southward along both the volcanic front and the Mariana Trough (i.e., from 25° to 21° and beyond) and northward to 26°. The authors need to explain why their observations cannot be explained just by differences in what is subducted at those latitudes, or how much is added to the mantle at various distances behind the volcanic front or above the slab. That is, the slab component may have always differed in kind or amount or ubiquity between at 24°, 21° and 18°N.

Response: Thank you for your comment. We agree with you that subduction of different components at different latitudes may produce different geochemical signatures. However, if we look at the arc whole rock data, most AVP, NSP, and CIP lavas have similar Pb isotopic compositions and all fall in or near the FOZO region (Fig. 3a in revised version). This indicates that these mantle source regions may be affected by similar proportions (~0.5%) of HIMU components, assuming that they all have the same Philippine Sea MORB mantle as the dominant source. In addition, subduction

components should be similar in such a small area as the NSP-24 source region, but only the high-Fo (>88) olivine hosted melt inclusions have obvious HIMU characteristics (Supplementary Table 3). This may be due to a small degree of partial melting during rifting producing a primitive melt with clear HIMU characteristics. The Mariana Trough at 18°N may not be affected by HIMU components because of the distance from the trench; alternatively, HIMU components in the mantle source were consumed with increased partial melting needed to create MORB-type magma beneath the back-arc basin. In the revised version, we emphasize that different regions are affected by different types and extents of subducted slab components (lines 255-324). In addition, we have revised the introduction section to emphasize the differences in subduction components in different regions of the IBM arc (lines 49-68), and highlight the unique geological background and the advantages of melt inclusions research for understanding melt formation beneath the NSP-24 region (lines 69-94). We hope you will be satisfied with our revisions.

The rift-to-spread argument also begs the question of when and how a HIMU-type component got as far south as 18°N beneath the MT where there is no slab currently. Did appropriate seamounts get subducted there too when the slab dip was shallower? Or did the slab-contaminated mantle flow south as the arc unzipped? Or was the HIMU character inherited from the Oki-Daito plume under the Philippine Sea Plate (Ishizuka et al., 2022) and is unrelated to subduction? Answering this may require careful comparison of the enriched components in the Mariana Trough and arc in light of recent data.

Response: Thank you for your questions and comments. In the revised version, we suggest that there is no HIMU component in the mantle at 18°N beneath the Mariana Trough, and no HIMU signal is reflected in trace elemental and Pb isotopic compositions (lines 379-382). This may be because the Mariana Trough at 18°N is located far away from the trench and no subducted seamount material exists beneath here, but this area is influenced by subducted sediments as indicated by its trace element characteristics and some element ratios (e.g., Ba/Nb>7). In addition, the nearby Oki-Daito mantle plume does not have very low $^{207}\text{Pb}/^{206}\text{Pb}$ (0.83-0.86) and $^{208}\text{Pb}/^{206}\text{Pb}$ (2.05-2.11) ratios (Ishizuka et al., 2013), so the HIMU features in NSP-24 melt inclusions may not be inherited from the Oki-Daito mantle plume.

Ishizuka, O., Taylor, R. N., Ohara, Y., & Yuasa, M. (2013). Upwelling, rifting and age-progressive magmatism from the Oki-Daito mantle plume. Geology, 41(9), 1011–1014.

The NSP24 host rocks and their OHMI are very similar to the Hiyoshis with the principal difference being their lower Ba/La, Pb/Ce, and Sr/Nd ratios despite higher La/Yb. Those differences may reflect their reararc setting, such as in the Kasuga and Guguan cross-chain (Stern et al., 1993, 2006). All three samples are from the eastern scarp of the remnant arc, ~40 km behind the volcanic front. They are 2.5-3.3 Ma and may pre-date rifting. At best, they are from the earliest stage of extension. I agree with the authors that their mantle source probably underlay the arc just before, or as, it started to rift, but these are samples of the reararc 3 m.y. ago, not the rift now.

In light of this, I suggest that the paper be re-structured to focus more on the differences in slab components and less on the rift-to-spread process. If the authors retain an emphasis on the latter, then they must address other places where that process can be, and has been, studied more thoroughly than here. These are primarily in the southern hemisphere including the Manus, North Fiji, and Lau Basins, and Kermadec Trough. Spreading at the East Lau and Futuna Spreading Centers in the Lau Basin reaches as close to the volcanic front as the WMR is to Hiyoshi. If anything,

the nature of the slab component and its effects on mantle melting during initial rifting can be studied there even better than in this paper (e.g., Bezos et al., 2009; Escrig et al., 2012). The rifting stage is known in the greatest detail in the Havre Trough (e.g., Gill et al., 2021). The current paper is provincial in this respect.

Response: Thank you for your suggestion, which we have adopted. In the revised version, we have reduced the discussion of rifting-to-spreading process. In the introduction section, we have removed some description of rifting-to-spreading process and re-emphasized the differences in slab components in the mantle source in different regions (lines 49-76). The NSP-24 region predates or began close to the time of rift propagation, providing an excellent opportunity for us to understand mantle source compositions during rifting. Our study does not focus on the mantle composition during rifting but on or before the earliest stages of extension to rift formation (i.e., initial arc rifting). We have changed the title to “A HIMU-like component in Mariana Convergent Margin magma sources during initial arc rifting revealed by melt inclusions”. In this region, the evolution from island arc to rifting occurred, the crust thinned, and the mantle composition should be more complex than that of mature island arc and back-arc basin. In addition, this region can be compared with back-arc basin and island arc activity in neighboring areas. They are a continuous transition of different stages of evolution of an island arc. This makes NSP-24 an ideal region for melt inclusion studies to identify complex components in the mantle source.

Second, the OHMI are less novel than claimed. The title proclaims that a HIMU-like has been revealed by their melt inclusions, but it was discovered almost 60 years ago by Tatsumoto at Io-To, then called Iwo Jima, in one of the first papers about Pb isotopes in volcanic rocks! The Pb isotopes in the Group 2 NSP24 OHMI are similar to those in Io-To (Tatsumoto, 1966; Sun and Stern, 2001) and Kita-Iwojima (Ishizuka et al., 2010) even though the major and trace elements differ. The opposite is true for the Group 2 NSP21 OHMI that are similar to the high-K basalts of the nearby Kasuga 2 and 3 seamounts in everything except their weird Pb isotopes and low chalcophile element contents.

Response: Thank you for your comment. We agree with you, but both Iwo Jima and Kita-Iwo Jima areas are located about 200 km north of our study area, and belong to the Volcano Arc (VA) region. The NSP-24 study area is close to the north Northern Seamount Province (N-NSP), is in the stage just prior to initial rifting, and thus has different tectonic setting than the Volcano Arc. In addition, there is no significant HIMU components in West Mariana Ridge whole-rock data near the N-NSP. The obvious HIMU nature of NSP-24 melt inclusions is reported for the first time here, indicating that HIMU components also exist in the mantle this far south. Compared with andesitic-dacitic volcanic rocks in Iwo Jima and Kita-Iwo Jima, the melt inclusion data are primitive enough to approximate mantle source compositions. The high-K and Pb isotope anomaly components of NSP-21 were also found for the first time in the volcanic front near 21°N, as a result of melt inclusion studies. Therefore, we believe that the study of melt inclusion in Mariana arc and Trough updates our previous understanding that some subduction components exist on a larger scale than indicated by whole rock studies, which is one of the novelties of our study. In addition, this study focuses on the melt inclusion characteristics before or during the earliest stage of arc rifting, which provides support for our understanding of the mantle composition under this special geological background. Third, the paper over-states the importance, and over-interprets the origin, of the weird Group 2 OHMI at NSP21 that are a new discovery in this paper. By “weird” I mean their low chalcophile element contents and their Pb isotope ratios. The host rock (T1-3) is similar to the high-K basalts at

the nearby Kasuga 2 and 3 seamounts (Fryer et al, 1997). The OHMI are similar in MgO and some other oxides to the dacites from Kasuga 3. It seems like special pleading to infer an entirely different slab source for these 3 wt% MgO OHMI in Fo70 olivine when so much is similar to nearby rocks. Why not appeal to something that just affects chalcophile elements during differentiation? For example, sulfides are known to fractionate Pb isotopes such that coexisting pyrite-sphalerites can have at least the range of Pb isotopes as in these OHMI (Yuan et al., 2015: Science China: Earth Sciences). After all, the Sr-Nd-Pb-Mo isotopes all vary considerably as a function of MgO at this location (Li et al., 2021), so it is an open system. Adding an exotic hydrothermal fluid to differentiated, Pb-poor melt might be an easier explanation than the 80 lines of speculation that start at l.331.

Response: Thank you for your suggestion. In the revised version, we have simplified the discussion of the source of NSP-21 K-rich magma (lines 285-306). We think that the fractional crystallization of sulfide can control the element contents of the melt and the assimilation of differentiated, Pb-poor and K-rich slab-derived components can explain the isotopic variations of the NSP-21 K-rich melt inclusions.

Moreover, whatever their origin, they are a tiny fraction of the OHMI that were studied, and they can be only a small mass fraction of magma at the volcanic front including their host rock. Try mass-balancing the OHMI to equal the host rock. The claim that this component “dominates the Mariana subarc mantle” (l. 542) is a vast over-statement. These are an intriguing discovery on the flanks of one seamount, but there is no evidence that they are a common component of Mariana volcanic front lavas.

Response: Thank you for your comment. We agree with you and have removed this sentence in the revised version. We believe that some HIMU signals in the mantle source may be obscured by different types of magmas and modified by late magmatic evolution, resulting in the absence of a clear HIMU signal in most whole rocks. On the contrary, under special geological circumstances, such as lithospheric thinning caused by rifting, the magmas derived from HIMU mantle sources that undergo a small degree of partial melting and may be captured by olivine, forming the melt inclusions in our study. This further indicates that magma in the initial rift stage has the potential to retain more complete information about mantle sources affected by subduction. We have changed “Our melt inclusion study demonstrates that the existence of a HIMU-like component in the oceanic arc settings may be more common than previously thought...” to “Our melt inclusion study demonstrates that early rift magma retain more complete information about subducted components and that a significant HIMU-like component in the Mariana subduction system may be more widespread than previously thought, at least to the north of 24°N,...” in lines 383-386.

An alternative emphasis for the paper might be how OHMI preserve slab component end-members at each site that are diluted in whole rocks (i.e., the subduction components section in l.289ff). To me, their most striking discovery is one that they do not discuss even though it is the best evidence for their claim that a wide range of slab components, including volcanoclastic as well as pelagic sediments, might be present beneath the Mariana Trough as far south as 18°N. The Pb in their OHMI from 18°N covers the full range of Pb isotopes of the entire Mariana Trough from 16-23°N, extending from the ambient mantle of Woodhead et al. (2005) to their most enriched MT basalts. There is even a negative correlation between $^{208}\text{Pb}/^{206}\text{Pb}$ ratios and Ba and Th-enrichment in both the OHMI and whole rocks. For NSP24, the OHMI data mean that they found the extreme HIMU

component as far south of Iwo-Tô as Ishizuka et al. (2007) found it to the north at Kita-Iwojima, and that it occurs even in the reararc where Pb/Ce ratios are lower, K₂O is higher, and the slab component is melt-like. This may be important but will require careful consideration of published data for all samples from 23°-26°N. As the authors already say, the NSP21 OHMI data mean that small amounts of a high-K component are present even at the volcanic front as well as behind it in the adjacent Kasuga cross-chain.

Response: Thank you for your suggestion. In the revised version, we have added the comparison and discussion of melt inclusion data with existing whole rock studies in the “subduction components” section (line 255), including Mariana Trough from 16-22°N (lines 277-282), Iwo Jima and Kita-Iwo Jima north to 25°N (lines 311-321), and Kasuga cross-chain near 21°N (lines 285-289). Our results show that olivine-hosted melt inclusions in the Mariana arc and Trough can reflect more end-member components, which is impossible to identify in whole-rock data, which is also one of the important results of melt inclusion research.

This paper’s discussion of the nature and origin of slab components is most persuasive about the HIMU component in L411ff, and that topic is in the paper’s title. The discussion seems reasonable to me, but I do not think that the data in this paper add new insight into the general origin of HIMU. It’s not like Zhang et al.’s (2022) discovery of high $\delta^{66}\text{Zn}$ values in HIMU lavas from the Cook-Austral islands and St. Helena. Consequently, while I found nothing wrong in that section, I do not think that it warrants such length. The overall model for NSP24 melts is essentially the same as in Ishizuka et al. (2010).

Response: Thank you for your suggestion. In the revised version, we have removed further discussion of the general origin of HIMU. HIMU signals in our OHMIs are more obvious than those in the whole rock studied by Ishizuka et al. (2010) in NSP-24, which is a key observation from our new melt inclusion data. In addition, a bulk components mixing model (mélange model) is used to simulate the source of NSP-24 melt inclusions.

In addition to those main points, here are some more specific things to address during revision.

The paper needs much greater clarity about exactly what samples were studied.

- MT18. Are the analyses of L1 and L3 in Supp.Table2 new analyses of the samples by that name in Zhao et al. (2016) that in turn came from Tian et al. (2005)? If so, then by what analytical methods? The data differ between this paper and Zhao/Tian by almost 2 wt% MgO and 25 relative% Na₂O, for example, and by a factor of 4 in some trace elements. Or, are these new analyses of the same or similar samples from 18°12’N that have a different name (T2-1,2,3) in another recent paper about Mo isotopes by the same first author of this paper (Li et al., 2021:Geology)? Give the lat/long and water depth at which the samples in this study were collected, details of their collection, and how they compare to other samples from the site. Then discuss how these two samples relate to the heterogeneity known along the MT between 18-19°N (segment 6 of Pearce et al., 2005). Trace element and Pb isotope ratios can vary significantly in <1 km in this segment, and all the samples in the Li and Zhao/Tian papers plus those from the same area in Chen et al. (2021) are more enriched than many (cf. Pearce et al., 2005; Woodhead et al., 2015). Neither sample in this paper meets the criterion (Ba/Nb<7) for unmodified (ambient) mantle wedge of Woodhead et al. (2015) whose Pb isotopes plot about where the letters CMT are in Fig. 3a of this paper. That strengthens the authors’ inference that even their MT18 samples have a significant slab component but they chose an

especially enriched example to study. They also might note that everything north of 19.8°N -- where the seafloor fabric deteriorates at the north end of Pearce et al.'s (2005) segment 3, west of Asuncion and Maug at the volcanic front -- appears enriched in a slab component based on the recent results of Chen et al. (2021) using old SIO dredges. Whether that change reflects melting out of a uniform slab component during rifting-to-spreading, or a change in the amount or kind or ubiquity of a slab component, is the question.

•NSP21. In reply to questioning, the authors said that sample T1 in SuppTable2 is a new analysis of a sample like T1-3 in Li et al. (2021), and they provided a map showing its location between Eifuku and Kasuga on the volcanic front. That reference and map need to be in the paper. Also, it should be noted that Eifuku has the lowest Nd and Hf isotopes south of the NSP, consistent with the addition of the most sediment (mixed pelagic-volcaniclastic) to its mantle source, according to Woodhead et al. (2015). The sample from this location with the lowest Nd isotope ratios in Li et al. (2021) also has the lightest Mo, consistent with that information.

•NSP24. In reply to questioning, the authors said these samples are from the West Mariana Ridge (Ishizuka et al., 2010) including D13 that was not discussed in that paper. Again, this information and map need to be in the paper, and the map should identify the Hiyoshis.

Response: Thank you for your comments and suggestions on our sample information. L1 and L3 have the same sampling information as Zhao et al. (2016). We acknowledge that in our previous version we focused on the data quality of melt inclusions and neglected to review the whole rock data. In order to more accurately characterize the geochemical composition of the whole rocks, we re-analyzed MT-18 and NSP-24 whole rock samples during the revision of the manuscript. During the analysis of the whole rocks, we conducted a rigorous review of the testing process and standard sample detection, and added duplicate samples for monitoring precision. The new whole-rock data are presented in Supplementary Table 2, and analysis methods are described in the Supplementary text. The gap between the new analytical data and the previous data may be due to the low accuracy of older analyses (nearly 20 years) and/or due to sample heterogeneity. In addition, the wide variability in different melt inclusions in the same sample also indicates that the MT-18 mantle has small-scale heterogeneity, similar to that reported in Pearce et al. (2005), which may be caused by subduction of different slab components, as you mentioned. Compared to other samples at the same location, our MT-18 samples may be enriched in subducted slab components due to their high Ba/Nb values. However, no matter which component (lithosphere, deep-subduction, shallow-subduction) is enriched, it does not show any HIMU signature, indicating that there was no HIMU material present beneath the Mariana back-arc region or that the HIMU component was strongly diluted. Chen et al. (2021) discussed the variation of mantle source compositions from north to south in the Mariana Trough and shown that back-arc spreading center magmatism is characterized by increasing asthenospheric mantle melting, decreasing slab material contribution and diminishing sub-arc lithospheric mantle participation. This is outside the scope of our study because we only studied Mariana Trough samples from 18° N. In this paper, we use different melt inclusion compositions in different convergent margin tectonic settings (arc, rift, back-arc basin) to reflect their different mantle sources. When plotting whole rock data, we collected all published data for igneous rocks of the study area, which span a compositional range covering our whole rock samples and indicating that our new whole rock compositional data is credible. In the revised version, we added a detailed map (Fig. 1) and brief description of samples (Supplementary Table 1) and related references. We hope you will be satisfied with our revisions.

*Pearce, J. A., Stern, R. J., Bloomer, S. H., & Fryer, P. Geochemical mapping of the Mariana arc-basin system: Implications for the nature and distribution of subduction components. *Geochem. Geophys. Geosyst.* 6(7) (2005).*

*Chen, Y., Niu, Y., Xue, Q., Gao, Y. & Castillo, P. An iron isotope perspective on back-arc basin development: Messages from Mariana Trough basalts. *Earth Planet. Sci. Lett.* 572, 117133 (2021).*

l.59. Few if any MT basalts cannot be distinguished from MORB using criteria like H₂O/Ce or Ba/Nb. Even Th/Yb vs Nb/Yb (Fig. 4a) distinguishes most.

Response: Thank you for your comment. In the revised version, we have removed this sentence. According to your suggestion, we have made extensive changes to the introduction section, deleted some descriptions about rifting-to-spreading process, and explained that different IBM arc segments are differently affected by subduction components.

l.68. What does “typical” mean? That requires comparison to other examples like those in the southern hemisphere. This is one example of un-necessary over-reach because of emphasizing the process of backarc basin evolution.

Response: We have removed this sentence in the revised version.

l.79. Why is the N-NSP “ideal”? As noted above, the chosen samples may even pre-date rifting.

Response: In the revised version, we have changed “The north Northern Seamount Province (N-NSP) between 23°N and 24°N is immediately north of where the Mariana Trough terminates (Fig. 1) and is an ideal area for studying how back-arc extension begins and evolves. This region is undergoing back-arc rifting that is propagating northward...” to “The north Northern Seamount Province (N-NSP) of the Mariana arc between 23°N and 24°N is immediately north of where the Mariana Trough terminates (Fig. 1) and is undergoing northward-propagating back-arc rifting” in lines 69-71.

l. 173. Why “arc” tholeiites?

Response: We have removed “arc” in this sentence (line 166).

l.186. The OHMI are NOT more enriched in K₂O/TiO₂ than E-MORB, but they are enriched in Ba/Nb when MgO <7 wt%. The latter is how Woodhead et al. (2015) distinguished their source as having a slab component. Most also are enriched in Th relative to ambient mantle, although it is important to compare them to “Indian”-type MORB.

Response: Thank you for you comment. We have changed “enriched K and Pb” to “enriched Pb” in this sentence (line 179). In addition, in the spider diagram, the enrichment of Ba/Nb cannot be shown, so we added the description of enrichment of Ba/Nb and Th in the discussion section about subduction components (lines 275-277).

l. 261ff. Consider breaking this long paragraph into two, with the first being about fractional crystallization, and the second starting at l.266 about partial melting. For NSP24, the latter depends on Supp.Fig.6 that in turn depends on the oddly low HREE in all the NSP24 OHMI that is rare in arc rocks. That accounts for the high Sm/Yb despite low La/Sm in these OHMI. The authors attribute that atypicality to 20% residual garnet in a depleted peridotite mantle source. That much garnet is unlikely even for fertile DM. Moreover, it seems impossible to mass balance the heavy REE in these OHMI relative to their host or other nearby rocks. Therefore, the model in Supp.Fig.6 is unconvincing. If not an analytical artifact, then might such low HREE result from adding eclogite melt from the slab to the reararc mantle?

Response: Thank you for your suggestion. We have broken this paragraph into two, with the first being about fractional crystallization (lines 226-233), and the second about partial melting (lines 234-254). In the revised version, the mineral proportions of garnet lherzolite were re-selected ($Ol_{0.600} + Opx_{0.200} + Cpx_{0.100} + Gt_{0.100}$) and the degree of partial melting was simulated (Supplementary Fig. 7). The results show that the NSP-24 high-K melt inclusions have less degree of partial melting and more garnet residue than those in NSP-21 and NSP-24 medium-K melt inclusions. For the HREE, we re-checked the data, and the analysis results of the reference samples were relatively good, with a relative error of less than 10% (Supplementary Table 4). The ranges of HREE in NSP-24 melt inclusions are large, but their distribution patterns are similar to that of host whole rocks. The lower HREE content may be due to differences in the degree of partial melting, or to the addition of eclogite melt from the slab. However, due to the significant difference in element distribution patterns with adakites, we think the addition degree of eclogite melt should not be high, and it is difficult to change our simulation results of partial melting.

1.288. By “heterogeneous sources” do you mean a variable addition of slab component to a homogeneous ambient mantle? That seems like the simplest, and most conventional, option.

Response: Yes, we exclude the influence of magmatic processes such as fractional crystallization and partial melting. Based on the differences in element abundances and Pb isotopes of melt inclusions from different groups, we believe that these inclusions come from an originally homogeneous mantle that was made heterogeneous by addition of different subduction components. We have changed “heterogeneous sources” to “sources affected by variable subduction components” in line 252.

1.293. Why not say silicate melts instead of silicic fluids?

Response: Thank you for your suggestion. We have changed “silicic fluids” to “silicate melts” in this sentence (line 259).

1.302. Why not say that Th is less soluble than REE and HFSE in “fluid”, but then cite experimental evidence rather than inferences from other field studies.

Response: Thank you for your suggestion. We have changed “Th is insoluble relative to REEs and HFSEs in slab-derived fluids” to “Th is less soluble than REEs and HFSEs in slab-derived fluids” and cited Brenan et al. (1995) and Kessel et al. (2005) in lines 267-269.

1.312. Wouldn't adding the same mass of sediment melt to variably depleted mantle have the same effect?

Response: This sentence expresses that the contribution of sediments in subduction components is greater than that of the altered oceanic crust. In the simulation calculation, we assume that NSP-21 and NSP-24 have similar depleted mantle composition, that is, Indian Ocean type mantle. The difference between the compositions of NSP-21 and NSP-24 melt inclusions is mainly controlled by subduction component types and partial melting degree.

1.315-318. I haven't chased down all these references, but I didn't think that the first three appealed to a sediment melt component in MTB. The classic Stolper and Newman (1994) paper was about a low-density fluid, which this paper's authors seem to prefer too.

Response: The subduction components have less influence on the MT-18 magma source compared to contributions to the NSP magma source. We have revised these descriptions in lines 273-282.

1.340. What differs from the high-K basalts at Kasuga 2 and 3? The Pb isotope ratios are higher and chalcophile element contents are lower, but what about everything else?

Response: We would like to emphasize that the samples of NSP-21 have a different geological

background from the samples of Kasuga 2 and 3, with NSP-21 located in the arc front and Kasuga in the Cross-Chain extending to the back arc. In the revised version, we have changed “NSP-21 K-rich melt inclusions have especially high alkali contents and can be classified as alkaline series (Supplementary Fig. 2a), which are the first reported alkaline compositions in S-NSP and different from the high-K melts of Kasuga 2 and 3 seamounts from the Kasuga Cross-Chain in the northern part of the Mariana back-arc basin” to “NSP-21 K-rich melt inclusions have especially high alkali contents and can be classified as alkaline series (Supplementary Fig. 2a), similar to the high-K melts of Kasuga 2 and 3 seamounts from the Kasuga Cross-Chain in the northern Mariana back-arc basin” in lines 285-288.

1.411. NSP21 and 24 rocks have quite different Nd and Hf isotope ratios so they cannot have similar sources even if their Pb is similar.

Response: Thank you for your comment. We have removed “suggesting similar mantle sources” in this sentence (line 310).

1.420. The Pb isotopes of the high-K OHMI aren't then same as in the Hiyoshis or WMR, but they are like Iwo-Jima, which requires a de-coupling of Pb from K between those sites, which is cool.

Response: We have added comparison with Pb isotopes of Iwo-Jima in lines 311-321.

1.478. This model for the HIMU component is essentially the same as in Ishizuka et al. (2010) and seems reasonable. However, please also apply it to the Group 1 OHMI in the same sample. Presumably they have less HIMU (carbonatite?), but why then do they have lower Nb/Yb and higher Ba/La and Th/Nb than Group 2?

Response: Thank you for your suggestion. In the revised version, simulation calculations are also applied to the group 1 melt inclusions in NSP-24 (Fig. 3a). We believe that these two groups of NSP-24 melt inclusions have similar proportion of depleted mantle, subducted sediment and altered oceanic crust because they come from the same region. The difference between these two groups of melt inclusions lies in the different proportion of the HIMU components. In the NSP-24 group 1 melt inclusions, HIMU components are negligible, suggesting a higher proportion of subducted sediments and altered oceanic crust. This also explains their higher Ba/La and Th/Nb and lower Nb/Yb ratios compared to NSP-24 group 2 melt inclusions. In addition, the difference of the partial melting degree may also control the contents and ratios of trace elements in different groups of melt inclusions.

Tables:

- The uncorrected K₂O and P₂O₅ entries for Type 2 NSP21 inclusions are transposed in Supp.Table1.

Response: Thank you for your comment. We have revised it.

- I strongly urge the authors to obtain and use IGSN numbers for their samples prior to publication. Otherwise, as more data are published for these and other samples, it will be come increasingly difficult to keep track of what is being studied using just a D- or T-number.

Response: Thank you for your suggestion. We fully agree with your suggestion. I have obtained 6 IGSN numbers of the samples, which are 10.58052/IE9080001 to 10.58052/IE9080006, and the data have been published on EarthChem Library (Xiaohui, L., 2024. Melt inclusion data from the Mariana subduction zone, Version 1.0. Interdisciplinary Earth Data Alliance (IEDA). <https://doi.org/10.60520/IEDA/113107>).

- Is there a rationale for the sequence of samples numbers in Supp.Table1?

- > Specify the host rock of the melt inclusions in MT18 and NSP24, and group the OHMI in each by host rock.
- > If any of the OHMI are from the same olivine, then group them together too.
- > Beyond those two principles, why not present the data in order of decreasing Mg# of the host or the melt inclusion? That would help readers to see the data in a geochemical context.

Response: Thank you for your suggestion. We have modified the tables according to your suggestion.

Figures:

- Figure 1 or a Supplementary equivalent needs to have much more detailed maps showing the location of the samples; something like Fig. 1c of Ishizuka et al. (2010) for each site.

Response: In the revised version, we have added detailed maps (Fig. 1) showing the sample locations.

- Figure 2. Perhaps as Supplemental equivalents, I wanted to see a least-squares best fit to mass balance between OHMI and host rock for c+d, and how the OHMI compared to whole rock compositions for the Hiyoshis and Iwo-To (Iwo-Jima) for e+f.

Response: In the revised version, we have added whole-rock data of these regions for comparison and described them in the figure caption (Supplementary Fig. 5).

- Figure 3a. This is such a small scale and has so much white space that it is hard to see details. Consider reducing the x-axis to 0.80-0.88 and then showing HIMU and EM1 in an inset because the data are more important than the model. I assume that the dashed yellow line is for NSP, but is there also a buried dashed red line for CIP, and is the white dashed line for CMT? The AVP, Iwo-Jima, and Kita-Iwojima, need to be added. All this should be explained in the caption. Why is the MORB source placed outside even the Pacific field? Why not place it inside the Indian field since the MT ambient mantle is Indian-type?

Response: Thank you for your suggestion. We have modified the Figure 3a to make it look clearer. We also have added the fields of AVP, Iwo-Jima, and Kita-Iwo Jima. In addition, we have changed the DMM end-member to Indian Ocean-type mantle field.

Reviewer #2 (Remarks to the Author):

review of Li et al: A HIMU-like component in the magma source during initial arc rifting revealed by melt inclusions

This manuscript presents new major and trace element concentrations and Pb isotope compositions for melt inclusions from lavas collected in three different regions of the Mariana arc: the mature back-arc, the northern portion of the main arc front, and the northern edge of the back-arc (NSP-24) where rifting is taking place. The authors show that all three locations exhibit distinct chemical and Pb isotopic compositions, largely confirming previous bulk rock studies from the same regions. The main difference to previous bulk rock work is that a group of melt inclusions from the NSP-24 region exhibit anomalous Pb isotope ratios that bear some similarities to so-called HIMU ocean island basalt lavas. After considering different scenarios of the HIMU origin, the authors conclude that it most likely represents seamounts that were subducted together with the Pacific plate. As such, no HIMU mantle source is present in the sub-arc Mariana mantle wedge, it is simply carried by the incoming subducted slab.

The quality of the data is good and, overall, I agree with the authors that their general interpretation is the most likely to account for a HIMU source in subduction-related lavas. In general, the paper certainly is worthy of eventual publication, but I am less sure of just how novel the paper really is given that the HIMU story feels more like an addendum to the generally well-studied Mariana arc system. I suppose it is up to the editor to determine if that merits publication in Nature Communications.

Response: We are grateful for your professional and constructive comments that helped us strengthen our arguments. We have revised the manuscript according to your comments and suggestions. In the revised version, we use melt inclusions to study magmatic source compositions at the initial stage of back-arc rifting. These NSP-24 high-K melt inclusions are primitive enough (MgO ~10 wt.%) to be directly representative of primary mantle melts. These data indicate the presence of extreme HIMU components that were not identified by previous whole-rock studies in this region, expanding our understanding of slab components, especially seamount subduction. This highlights the influence of seamount subduction on arc magma compositions, which is often overlooked in conventional studies. In addition, we present the compositional characteristics of melt inclusions in different tectonic evolution stages, and these data reflect different contributions of subduction components to the mantle source in three different tectonic settings of the Mariana system, which provides a new perspective for us to understand convergent margin magma generation. We believe that these studies are novel and worthy of publication in Nature Communications. We also hope you are satisfied with our explanations and efforts in the manuscript revision.

One major problem I had while reading this paper was that all the figures were presented in low resolution with very small font sizes. I, therefore, really struggled to see some of the details of the plots. I would urge the authors to fix this and recommend the Nature Communications do not allow papers with almost illegible figures to go out for review.

Response: Thank you for your suggestion. We have made substantial modification to all the figures to improve their resolution and readability.

When diving into a more detailed view of the manuscript as written I have some issues, both with

the way the interpretations are presented as well as the processes that are implied to drive the subduction processes. I outline these in more detail below:

1. The paper is far too long and some of the interpretations, while I generally agree with them, are argued in a convoluted way. For example, the section on the origin of the NSP-21 K rich lavas is almost four pages long. Yet, the authors end up concluding that the origin of these lavas is mostly due to assimilation-fractional-crystallization (AFC) type processes. I think this makes sense, but the authors could likely have stated this in less than a page. Instead of providing a long-winded discussion of other processes, I don't understand why they didn't just go straight to the strong evidence for their conclusion (low chalcophiles, low MgO, correlations between Pb isotopes and MgO, and more). Supplementary figures 3, 4, 5 do a great job of documenting all this.

Response: Thank you for your suggestion. In the revised version, we have simplified the discussion of the source of NSP-21 K-rich magma (lines 285-306), which makes our manuscript more concise and readable. The generation of these K-rich magmas is mainly attributed to the contamination of subducted K-rich and sulfides separated slab components during magma evolution.

Another example is the very long discussion the authors present on the origin of HIMU mantle sources. This includes a detailed account of arguments supporting carbonate involvement in HIMU generation. However, since the main conclusion of the paper is that the HIMU signature comes from a seamount, I do not see any reason to even mention carbonates or carbonatites. The only evidence the authors present that they claim support a carbonatite involvement in the HIMU source is figure 5, which I find very tenuous. While the trace element ratios shown there are distinct from the other melt inclusion groups, they are only very slightly offset towards a carbonatite source and what they fail to mention is that the observed values fully overlap with normal mantle values. Hence, there is nothing particularly 'carbonatite-like' about the observed trace element data. In the end, none of that matters for the interpretation of the origin of the HIMU signature in the NSP-24 samples. It makes sense that it is a subducted seamount and whether that seamount had a carbonatite or an altered oceanic crust influenced mantle source is irrelevant.

Response: Thank you for your suggestion. In the revised version, we have removed the discussion of involvement in HIMU generation and also removed the Figure 5 (Zr/Hf vs. Ce/Pb and Sm/Yb vs. Ce/Pb). After modification, we emphasize that the HIMU signal is caused by seamount subduction (lines 326-351).

2. There are some inconsistencies between the processes the authors seem to infer occurring during subduction and what has been proposed in the literature. For example, the authors claim that the mantle wedge is metasomatized by subducted sediments. To me it looks like these sediments are simply just bulk sediments. But typically, studies have suggested that sediments will experience partial melting before entering the mantle wedge and resulting in metasomatism. If the authors want to pursue such an interpretation, they would need to use sediment melts calculated based on one of the available experimental studies that have investigated such a process (eg Hermann and Rubatto, 2009). Including this calculation will primarily affect the trace element ratios of the resulting mantle source as well as the proportions of sediment required to explain the Pb isotopes.

Furthermore, I find it difficult to reconcile the HIMU seamount as a bulk component in the mixing calculations because that would require high degrees of partial melting of what is essentially basaltic oceanic crust. I am very doubtful about such a partial melting process since oceanic crust becomes eclogitic at sub-arc depths. In turn, melting of eclogite only happens at very elevated temperatures

and also results in very specific geochemical signatures, such as high Sr/Y ratios (eg adakites). As far as I can tell, this is not observed in the studied samples.

If the authors prefer to consider bulk sediment and oceanic crust mixing processes for their Pb isotope and trace element calculations, then they are basically inferring so-called melange formation (eg Marschall and Schumacher 2012; Nielsen and Marschall 2017). Instead of the long-winded discussions of the NSP-21 samples and the origin of HIMU, it might make more sense to present a discussion of different subduction processes like mantle metasomatism and melange formation. To me that would be more directly related to the presented data set and would also, in my opinion, make the paper of broader more general interest that might be better suited to Nature Communications.

Response: Thank you for your suggestion. We fully agree with you that it makes sense to clarify the transfer process of subducted slab materials to mantle wedge. However, we do not know the details of how the HIMU-like component was transferred from the subducting plate to the zone of melt generation; this is an important question but it is not the major point of our study. The focus of this study is to reveal the composition of the mantle source region in the initial arc rifting stage of the Mariana subduction zone, and to clarify the storage mechanism of this HIMU-like component under this special background. Only the melt inclusions hosted by primitive olivine show HIMU-like characteristics, while the whole rocks in this region do not. In fact, both the metasomatized mantle-wedge melting model and the *mélange*-diapir melting model can transfer the subducted HIMU components to the over-riding plate, so the melt transformation process will not affect the overall logic of this paper, and does not change the fact that HIMU components are present in the Mariana island arc source region. Nevertheless, in order to facilitate the calculation, in the revised version, we chose the bulk components mixing model, that is, *mélange* model, to draw the Figure 3a. If the metasomatic model is used to calculate, only the proportion of end-element components changes and the overall conclusion does not change. In addition, we have removed the description of metasomatism in the revised manuscript. We hope you will be satisfied with our modification.

I also have some more specific line-by-line comments that the authors should address prior to resubmitting a revised version:

Line 108: I looks like the latitudes are wrong for the SMT

Response: Thank you for your comment. We have changed “20.5-17.5°N” to “12.5-17.5°N” in line 105.

Line 155: what about re-equilibration? You just mention it here but do not further explore it. This process is the only one that is likely to affect isotopes and incompatible trace element ratios, so that seems more important to discuss than the PEC

Response: As described by Danyushevsky et al. (2000), the re-equilibration process, referred to as “Fe-loss”, can operate during natural pre-eruptive cooling of host magma and results in lower FeO_t and higher MgO contents within the initially trapped volume of inclusion. This process can be corrected using the Petrolog 3 software. We have described the correction process in lines 148-159. Due to being captured by olivine, the re-equilibration process mainly affects MgO and FeO concentrations of melt inclusions, but has little effect on other major-element oxides and isotopes, thus the overall geochemical trends and composition changes of melt inclusions remain little or

unaffected.

Danyushevsky, L.V., Della-Pasqua, F.N. & Sokolov, S. Re-equilibration of melt inclusions trapped by magnesian olivine phenocrysts from subduction-related magmas: petrological implications. Contrib. Mineral. Petrol. 138, 68-83 (2000).

Line 194: delete significance

Response: We have deleted it.

Line 306-308: you should note that some sediments can have very large Ba enrichments with Ba/La and Ba/Th ratios that exceed those of arc lavas. Hence, without knowing the Ba contents of subducting sediments outboard of your arc lavas, you cannot necessarily attribute Ba enrichments to fluids (neither from oceanic crust nor from sediment). As shown in recent Ba isotope papers on arc lavas, there is little evidence to suggest that Ba in arc lavas is primarily supplied by ocean crust derived fluids.

Response: Thank you for your comment and suggestion. In the revised version, we have removed the plot of “Ba/La vs. Th/Yb” and “Ba/Th vs. Th/Nb” and added a plot of “Ba/Yb vs. Th/Yb”. We have described the high Th/Yb and Ba/Yb ratios as the contribution of subducted sediments (lines 271-273).

Line 310: I dont see this. Ba/La is quite variable, whereas Ba/Th is more constant. This may imply that Ba and Th are both coming from sediment and that fluids are less important in supplying Ba to the mantle source.

Response: Thank you for your comment. We agree with you and have changed “NSP-21 and NSP-24 melt inclusions have highly variable Th/Yb and Th/Nb ratios but relatively constant Ba/La and Ba/Th ratios, suggesting that the contributions of subducted sediments or sediment melts are more variable than those of subducted altered oceanic crust-derived fluids” to “NSP-21 and NSP-24 melt inclusions have highly variable Th/Yb and Ba/Yb ratios, suggesting that the contributions of subducted sediments are more variable than those of subducted AOC” in lines 271-273.

Line 315-317: I do not follow what evidence you have presented that supports this conclusion about the slab component in the mature back-arc. The subduction component is generally very small and, therefore, is difficult to determine the origin of.

Response: Thank you for your comment. We have removed the descriptions of the origin of subduction components and suggested that the subduction components have less influence on the MT-18 magma source compared to contributions to the NSP magma source.

Line 436-439: I dont think the paper by Zhang et al makes a strong case for carbonatite origin of HIMU. Zn isotopes are essentially unexplored in subduction zones and the paper presents no information about what might happen to Zn isotopes in a residual slab. So to me it is equally possible that small amounts of Zn isotope fractionation occurs during slab dehydration. I also cannot recognize the description of the Zn isotope values as 'exceptionally high' since they are only about 0.1 permil heavier than MORB and even less compared with other OIBs.

Response: We have removed the description of Zn isotopes research in HIMU origin in the revised version.

Line 441-443: I dont really agree that this is clear. The NSP-24 samples are only slightly plotting towards the carbonatite field and I dont see a clear reason for why carbonatite needs to be implicated.

Response: We have removed this discussion.

Line 436-455: I do not understand the reason for the authors wanting to push the argument of HIMU

being a carbonatite source. I don't think it has much to do with their data set.

Response: Thank you for your comment and suggestion. In the revised version, we have removed further discussion of carbonatite and HIMU sources.

Line 477: why use a generic marine sediment field instead of sediments outboard of the Mariana arc?

Response: In the revised version, we used the sediments from ODP 801 outboard the Mariana arc as the sediment end member for the simulation calculations (Supplementary Table 6).

Line 478-481: This text seems to imply bulk mixing processes were considered. As I note above, such a process is not compatible with mantle metasomatism but instead more reminiscent of mélange formation followed by partial melting.

Response: Thank you for your suggestion. We agree with your comments about mélange mixing. In the revised version, we have used a bulk components mixing model to simulate the calculation (lines 329-334), which is the mélange-diapir melting model and could reasonably explain our data.

Line 492-494: I still see no particular evidence to support that the HIMU component in the NSP has anything to do with carbonatite or carbonated mantle.

Response: We have removed the “characterized by carbonated mantle peridotite” in this sentence (line 365).

Line 509-527: I think the authors would be better served by deleting this whole paragraph.

Response: Thank you for your suggestion. We have removed this paragraph in the revised version.

Line 546-554: Alternatively, it is more of a random chance problem. There will not be HIMU seamounts subducting all the way along the arc. Maybe there is a HIMU seamount subducting under the NSP-24, but not under NSP-21? Also, the mature back arc samples only carry a very minor slab component, which is very difficult to identify the exact nature of because it only causes a small modification away from the MORB field. So that, in itself, might not have anything to do with a progressive depletion of the mantle wedge. In any case, the NSP-24 samples only reveal a quite dilute HIMU signature, which, when further strongly diluted, would be essentially impossible to distinguish from any other slab component.

Response: Thank you for your comment. We agree with you. However, for Mariana arc, there are abundant HIMU-seamounts distributed in the west Pacific outboard of the Mariana arcs. The Wake seamount chain and its volcanoclastic components near 21°N have HIMU characteristics and are subducting into the mantle together with the Pacific Plate. In addition, whole-rock data indicate that most Mariana arc volcanic rocks have FoZo-characteristic Pb isotopic compositions. This isotopic composition can be obtained by the injection of HIMU component in the mantle wedge. Therefore, we infer that more HIMU seamounts may be subducting than previously thought (lines 354-366). These HIMU signals may be diluted by subarc mantle magma and generally are not well presented. Under some special conditions, such as low degree partial melting of the mélange, the melt of the typical HIMU signal can be captured and form the NSP-24 high K melt inclusions. This means that melt inclusions can reveal more information about magmatic source composition than the whole rock. Hope you are satisfied with our response.

Line 556: I don't see how this reveals it is more common. It basically requires subduction of seamounts with a very specific HIMU composition. Whether these types of seamounts are more common than we think is another question. If yes, then it is possible that more arcs could include such a component.

Response: Thank you for your comment. We have changed “the existence of a HIMU-like

component in the oceanic arc settings may be more common” to “a significant HIMU-like component in the Mariana subduction system may be more widespread than previously thought, at least to the north of 24°N...” in this sentence (lines 384-386). As we answered in the previous question, the subduction of HIMU-type seamounts in the Mariana system may be more widespread than previously thought, and we have also described this in lines 354-366.

Line 634: the authors need to describe how they corrected or instrumental mass bias. And how variations in mass bias were accounted for.

Response: Thank you for your suggestion. We have added the relevant description in lines 449-451. The instrumental mass bias was corrected using the standard–sample–standard bracketing method and any outliers ($>2SD$) were excluded.

Line 634-641: Was the laser spot size the same for samples and reference materials? Based on the SE of individual melt inclusions and the reference materials, it looks like they must have been similar. However, there is a large number of melt inclusion measurements that have significantly higher SE counting statistics than the reference materials. So these likely have greater error bars than the 0.39% indicated for the 208/206 and 207/206 ratios and 1.22% indicated for ratios with 204 involved. I think the authors need to consider if some of their Pb isotope variations (especially within a specific group) might be due to analytical scatter. This is not the case for the variation between groups which is much larger than what can be attributed to analytical scatter.

Response: Thank you for your comment. We used the same laser beam spot size for the samples and reference glass. We have reviewed the data and believe it to be accurate and reliable. The analysis of reference glass is performed after every 8 melt inclusions, and the results show that the analysis accuracy of reference glass is excellent. Therefore, we believe that some Pb isotope variations in melt inclusions are due to their own characteristics rather than analytical scatter. In addition, when plotting, we added the error bars of the Pb isotopes, and the Pb isotopes of different groups of melt inclusions showed sufficient differences to allow comparison of end-member components (Fig. 3). We hope you will agree with our explanation.

REVIEWERS' COMMENTS

Reviewer #1 (Remarks to the Author):

The authors are to be congratulated and rewarded for their willingness and ability to revise their paper so thoroughly in light of reviewers' comments. At the same time, they significantly improved the quality of prose and figures in many places beyond reviewer input. Perhaps the co-authors engaged more this time. The result is a great improvement that could be accepted with only minor further revision. The improvements allow reviewers to focus on less central matters that could improve the manuscript further.

The description of samples needs further improvement either in Supplemental Information or Supplemental Table Sheet 1. How exactly are the samples in this study related to samples and data in previously published papers? This is discussed in the Rebuttal but I didn't find it in the revision. The IGSN numbers also need to be inserted in Supplemental Table Sheet 1, and the new ECL should be cited in the main manuscript.

One new idea introduced during revision is that the HIMU-like slab-derived component is restricted to the lithosphere and, therefore, evident only or mostly when the lithosphere melts during rifting. I see no reason not to make this suggestion, and the authors could cite Pearce et al. (2005) for support of a lithospheric component in the NSP due to high Nb/Ta ratios there. However, conceptually, why would a slab-derived HIMU component be restricted to the lithosphere? Instead, might it just be restricted spatially to where HIMU seamount chains are being subducted, thereby constricting backarc spreading?

The new figure 1 is a vast improvement over the original. It would have saved me many hours had it been submitted initially. The model in Fig. 3b is much improved, and the new Fig. 5 cartoon is lovely although it seems odd to show such thick lithosphere (if any) beneath the Mariana Trough after millions of years of spreading. L.380 even says that none needs to be present.

The authors missed the chance to estimate the mass fraction of melt inclusion types in their host rocks, allowing readers to dismiss the importance of their discovery of previously unknown melt compositions in these locations.

Line numbers below refer to the revised and merged document.

I.73: Consider change to "...erupted IN THIS LOCATION just prior to..." I know of no evidence that this difference occurs elsewhere.

I.89: Consider inserting "remnant arc" before West Mariana Ridge, to alert readers to the tectonic setting of this ridge earlier than I.120 as now. You could also replace "corresponding to" with "that is the conjugate pair of" to further emphasize the setting.

I.109: Consider removing "as a result of propagating rifting". This was added during revision but is a carry-over from the original focus of the paper. The authors provide no evidence that the different

proportions and types of subduction components in the mantle source result from rift propagation rather than, for example, what was subducted at different latitudes or times.

I.191: Consider replacing “and” with “from” because I think you mean strong relative LREE enrichment; i.e., fractionation of LREEs from HREEs.

I.196: This sentence falsely implies that there are five separate fields of Pb isotope ratios in Fig. 3 whereas there are only three. It might be better to say that “Each of the five types of melt inclusions defined above has a narrow range of Pb isotope compositions. The total range encompasses...”

I.244-246. I continue to find the model of melting garnet peridotite unconvincing and the HREE data suspect, but the authors’ rebuttal is reasonable. Variable subduction components, as discussed in the following section, seem a more likely explanation to me.

I.266. Use words instead of the PEC acronym.

I.272-273. Do you mean that the relative proportion of sediment and AOC in the subduction component is highly variable?

L277: Consider replacing “proposed” by “discussed” because the high Ba/Nb ratios were discovered and discussed, not proposed, in ref. 53.

I.302: Consider replacing “these experienced...” with “these melts were products of “ fractional crystallization.

I.336: Remove final “s” from patterns.

I.352ff: Although the model is reasonable, as usual there is ambiguity whether an intermediate FOZO-like source component would suffice without the leverage of a HIMU extreme. However, no need to revise.

I.357: Should “metabasites” be “metabasalts”?

I.380: This run-on sentence needs revision. Do you mean “This may be because the back-arc basin spreading center is located far away from the trench, and is mainly fed by a large amount of MORB-like magma. If so, then no lithospheric HIMU component is present, or it has been diluted.”

Reviewer #2 (Remarks to the Author):

I have had another read of the revised manuscript on melt inclusions from the Mariana arc, back-arc and northern rift zone. As outlined in my initial review, I found that the main conclusions were well supported by the data presented and my main criticism involved the length of the manuscript as well as a few conceptual modifications.

I think the authors have done a very nice job of addressing all my comments and suggestions. The paper is now much easier to read both due to shortening and tightening up of a few sections.

I have no major issues that I think need to be addressed. I have a single comment to line 205 where the authors state that the MT-18 lavas have Pb isotope compositions similar to Indian MORB. The authors might add here that the radiogenic isotope composition of the mantle wedge is very commonly observed to be similar to Indian MORB for almost all Western Pacific Subduction zones (Kamchatka, Kuriles, IBM, Ryukyu). See for example the following papers:

Pearce et al *J. Petrol.* 40, 1579–1611 (1999)

McCarthy et al *Geochimica et Cosmochimica Acta* 296, 170–188 (2021)

Martynov, et al *Island Arc* 19, 86–104 (2010)

Shu et al *Geochimica et Cosmochimica Acta* 217, 462-491 (2017)

Shu et al *Nat Commun* 13, 4467 (2022)

REVIEWERS' COMMENTS

Reviewer #1 (Remarks to the Author):

The authors are to be congratulated and rewarded for their willingness and ability to revise their paper so thoroughly in light of reviewers' comments. At the same time, they significantly improved the quality of prose and figures in many places beyond reviewer input. Perhaps the co-authors engaged more this time. The result is a great improvement that could be accepted with only minor further revision. The improvements allow reviewers to focus on less central matters that could improve the manuscript further.

Response: We appreciate your positive comments and important suggestions. We have revised them in our new submission.

The description of samples needs further improvement either in Supplemental Information or Supplemental Table Sheet 1. How exactly are the samples in this study related to samples and data in previously published papers? This is discussed in the Rebuttal but I didn't find it in the revision. The IGSN numbers also need to be inserted in Supplemental Table Sheet 1, and the new ECL should be cited in the main manuscript.

Response: Thank you for your suggestion. We have added a description of the differences between this study and the previous paper in the Supplemental Table Sheet 1. The IGSN numbers also have been inserted in Supplemental Table Sheet 1. The ECL has been cited in "Data availability" section in the manuscript.

One new idea introduced during revision is that the HIMU-like slab-derived component is restricted to the lithosphere and, therefore, evident only or mostly when the lithosphere melts during rifting. I see no reason not to make this suggestion, and the authors could cite Pearce et al. (2005) for support of a lithospheric component in the NSP due to high Nb/Ta ratios there. However, conceptually, why would a slab-derived HIMU component be restricted to the lithosphere? Instead, might it just be restricted spatially to where HIMU seamount chains are being subducted, thereby constricting backarc spreading?

Response: Thank you for your suggestion. We have cited this reference and added "high Nb/Ta ratios in NSP" in line 367. The density of the lithosphere is relatively small, no melting occurs, and this HIMU component can be retained in the lithosphere. In the asthenosphere, HIMU components may be diluted and not present. The occurrence of HIMU components is mainly due to the subduction of HIMU type seamount in NSP. However, samples from NSP-24 region do not all show the presence of HIMU components, which may be mainly due to the difference of melting degree in the process of lithosphere decompression.

The new figure 1 is a vast improvement over the original. It would have saved me many hours had it been submitted initially. The model in Fig. 3b is much improved, and the new Fig. 5 cartoon is lovely although it seems odd to show such thick lithosphere (if any) beneath the Mariana Trough after millions of years of spreading. L.380 even says that none needs to be present.

Response: Thank you for your positive comment. We have thinned the lithosphere in Figure 5b.

The authors missed the chance to estimate the mass fraction of melt inclusion types in their host rocks, allowing readers to dismiss the importance of their discovery of previously unknown melt compositions in these locations.

Response: Thank you for your comment. Indeed, it is difficult to give an exact proportion of HIMU-like melt at this stage. However, according to the data, it is clear that HIMU-like melt inclusions are mainly stored in olivine with high Fo values, which represents a sufficiently primitive melt component.

Line numbers below refer to the revised and merged document.

l.73: Consider change to "...erupted IN THIS LOCATION just prior to..." I know of no evidence that this difference occurs elsewhere.

Response: We have changed "... that magmas erupted just prior to initial arc rifting..." to "... that magmas erupted in this location just prior to initial arc rifting..." in line 65.

l.89: Consider inserting "remnant arc" before West Mariana Ridge, to alert readers to the tectonic setting of this ridge earlier than l.120 as now. You could also replace "corresponding to" with "that is the conjugate pair of" to further emphasize the setting.

Response: Thank you for your suggestion. We have changed "These include samples collected from the eastern scarp of the West Mariana Ridge (WMR) corresponding to the northernmost North Mariana Seamount Province around 24°N" to "These include samples collected from the eastern scarp of the remnant arc West Mariana Ridge (WMR) that is the conjugate pair of the northernmost North Mariana Seamount Province around 24°N" in lines 80-83.

l.109: Consider removing "as a result of propagating rifting". This was added during revision but is a carry-over from the original focus of the paper. The authors provide no evidence that the different proportions and types of subduction components in the mantle source result from rift propagation rather than, for example, what was subducted at different latitudes or times.

Response: Thank you for your suggestion. We have removed "as a result of propagating rifting" in this sentence (line 100).

l.191: Consider replacing "and" with "from" because I think you mean strong relative LREE enrichment; i.e., fractionation of LREEs from HREEs.

Response: Thank you for your suggestion. We have changed it in line 183.

l.196: This sentence falsely implies that there are five separate fields of Pb isotope ratios in Fig. 3 whereas there are only three. It might be better to say that "Each of the five types of melt inclusions defined above has a narrow range of Pb isotope compositions. The total range encompasses..."

Response: Thank you for your comment. We have changed "The 5 different groups of melt inclusions fall into different fields with little overlap (Fig. 3a), with a total range that encompasses ..." to "Each of the five types of melt inclusions defined above has a narrow range of Pb isotope compositions. The total range encompasses ..." in lines 188-190.

l.244-246. I continue to find the model of melting garnet peridotite unconvincing and the HREE data suspect, but the authors' rebuttal is reasonable. Variable subduction components, as discussed in the following section, seem a more likely explanation to me.

Response: Thank you for your comment. The description of this part only shows that partial melting will not significantly change the element composition of different groups of melt inclusions according to the simulation results. Therefore, the chosen melting model is not the main concern.

l.266. Use words instead of the PEC acronym.

Response: Thank you for your comment. We have changed it in lines 260-261.

l.272-273. Do you mean that the relative proportion of sediment and AOC in the subduction component is highly variable?

Response: Yes, what we mean is that the proportion of sediment contribution is variable.

L277: Consider replacing “proposed” by “discussed” because the high Ba/Nb ratios were discovered and discussed, not proposed, in ref. 53.

Response: Thank you for your suggestion. We have changed it in line 271.

L302: Consider replacing “these experienced...” with “these melts were products of “fractional crystallization.

Response: Thank you for your suggestion. We have changed it in line 296.

L336: Remove final “s” from patterns.

Response: Thank you for your comment. We have changed it in line 330.

L352ff: Although the model is reasonable, as usual there is ambiguity whether an intermediate FOZO-like source component would suffice without the leverage of a HIMU extreme. However, no need to revise.

Response: Thank you for your comment. We will consider this question.

L357: Should “metabasites” be “metabasalts”?

Response: Thank you for your comment. We have changed “metabasites” to “metabasalts” in line 351.

L380: This run-on sentence needs revision. Do you mean “This may be because the back-arc basin spreading center is located far away from the trench, and is mainly fed by a large amount of MORB-like magma. If so, then no lithospheric HIMU component is present, or it has been diluted.”

Response: Thank you for your suggestion. We have changed this sentence in lines 374-377.

Reviewer #2 (Remarks to the Author):

I have had another read of the revised manuscript on melt inclusions from the Mariana arc, back-arc and northern rift zone. As outlined in my initial review, I found that the main conclusions were well supported by the data presented and my main criticism involved the length of the manuscript as well as a few conceptual modifications.

I think the authors have done a very nice job of addressing all my comments and suggestions. The paper is now much easier to read both due to shortening and tightening up of a few sections.

I have no major issues that I think need to be addressed. I have a single comment to line 205 where the authors state that the MT-18 lavas have Pb isotope compositions similar to Indian MORB. The authors might add here that the radiogenic isotope composition of the mantle wedge is very commonly observed to be similar to Indian MORB for almost all Western Pacific Subduction zones (Kamchatka, Kuriles, IBM, Ryukyu). See for example the following papers:

Pearce et al *J. Petrol.* 40, 1579–1611 (1999)

McCarthy et al *Geochimica et Cosmochimica Acta* 296, 170–188 (2021)

Martynov, et al *Island Arc* 19, 86–104 (2010)

Shu et al *Geochimica et Cosmochimica Acta* 217, 462–491 (2017)

Shu et al *Nat Commun* 13, 4467 (2022)

Response: Thank you for your comment. We have added this sentence “Actually, the radiogenic isotope composition of the mantle wedge is very commonly observed to be similar to Indian Ocean-type mantle for almost all Western Pacific Subduction zones (e.g., Kamchatka, Kuriles, IBM, Ryukyu)” in lines 197-200.